EMBO
Molecular Medicine

# The cardiac microenvironment uses non-canonical WNT signaling to activate monocytes after myocardial infarction

Ingmar Sören Meyer[1,2], Andreas Jungmann[1], Christoph Dieterich[1,2], Min Zhang[1], Felix Lasitschka[3,4], Susann Werkmeister[1], Jan Haas[1,2], Oliver J Müller[1,2], Michael Boutros[2,5], Matthias Nahrendorf[6], Hugo A Katus[1,2], Stefan E Hardt[1] & Florian Leuschner[1,2,*] [iD]

## Abstract

A disturbed inflammatory response following myocardial infarction (MI) is associated with poor prognosis and increased tissue damage. Monocytes are key players in healing after MI, but little is known about the role of the cardiac niche in monocyte activation. This study investigated microenvironment-dependent changes in inflammatory monocytes after MI. RNA sequencing analysis of murine Ly6C[high] monocytes on day 3 after MI revealed differential regulation depending on location. Notably, the local environment strongly impacted components of the WNT signaling cascade. Analysis of WNT modulators revealed a strong upregulation of WNT Inhibitory Factor 1 (WIF1) in cardiomyocytes—but not fibroblasts or endothelial cells—upon hypoxia. Compared to wild-type (WT) littermates, WIF1 knockout mice showed severe adverse remodeling marked by increased scar size and reduced ejection fraction 4 weeks after MI. While FACS analysis on day 1 after MI revealed no differences in neutrophil numbers, the hearts of WIF1 knockouts contained significantly more inflammatory monocytes than hearts from WT animals. Next, we induced AAV-mediated cardiomyocyte-specific WIF1 overexpression, which attenuated the monocyte response and improved cardiac function after MI, as compared to control-AAV-treated animals. Finally, WIF1 overexpression in isolated cardiomyocytes limited the activation of non-canonical WNT signaling and led to reduced IL-1β and IL-6 expression in monocytes/macrophages. Taken together, we investigated the cardiac microenvironment's interaction with recruited monocytes after MI and identified a novel mechanism of monocyte activation. The local initiation of non-canonical WNT signaling shifts the accumulating myeloid cells toward a pro-inflammatory state and impacts healing after myocardial infarction.

**Keywords** inflammation; monocytes; myocardial infarction

**Subject Category** Cardiovascular System

## Introduction

Myocardial infarction (MI) is one of the leading causes of death in industrialized nations. Although improved treatment has tremendously increased survival of acute MI, the incidence of subsequent heart failure remains high (Burchfield et al, 2013). Our understanding of inflammatory processes following cardiac injury has progressed in recent years. We now know that an adequate inflammatory response is crucial for the healing process after cardiac injury, but prolonged and exaggerated (or diminished) inflammation can cause additional tissue damage and promote adverse remodeling (Saxena et al, 2016). Monocytes and monocyte-derived macrophages are critical players in not only the development of atherosclerosis and coronary heart disease but also the immune response to cardiac ischemia (Geissmann et al, 2010). Following acute coronary occlusion and subsequent myocardial damage, neutrophils are the first dominant leukocyte subset to invade the infarcted heart. Their numbers peak early, during the first days after cardiac injury, and monocytes and their lineage-descendant macrophages become the predominant infiltrating cell types over the course of the first week (Swirski & Nahrendorf, 2013). The monocyte/macrophage response is biphasic: Whereas pro-inflammatory Ly-6C[hi] monocytes dominate the early phase (1–4 days post-injury), reparative Ly-6C[lo] macrophages are the predominant cell type at later stages (Nahrendorf et al, 2007; Hilgendorf et al, 2014; He et al, 2015). The inflammatory Ly-6C[hi] monocytes are highly proteolytic and phagocytotic. They clear dead

1   Department of Medicine III, University of Heidelberg, Heidelberg, Germany
2   DZHK (German Centre for Cardiovascular Research), Partnersite, Heidelberg/Mannheim, Germany
3   Institute of Pathology, University of Heidelberg, Heidelberg, Germany
4   Tissue Bank of the National Center for Tumor Diseases (NCT), Heidelberg, Germany
5   Division Signaling and Functional Genomics, German Cancer Research Center (DKFZ) and Heidelberg University, Heidelberg, Germany
6   Center for Systems Biology, Massachusetts General Hospital and Harvard Medical School, Boston, MA, USA
    *Corresponding author. Tel: +49 6221 56 8676; E-mail: florian.leuschner@med.uni-heidelberg.de

cellular debris and damaged extracellular matrix from the infarcted area. The reparative Ly-6C$^{lo}$ monocytes/macrophages promote angiogenesis and collagen deposition to form a solid scar and replace the lost heart tissue (Swirski & Nahrendorf, 2013). Balancing monocytes/macrophages' pro- and anti-inflammatory properties seems to be a prerequisite for optimal cardiac healing. Going forward, we need further insights into the signals controlling this balanced response, especially since clinical anti-inflammatory strategies have failed thus far (Christia & Frangogiannis, 2013).

Recent research has revealed that WNT signaling has immunomodulating properties and is induced by inflammatory mediators (Pereira *et al*, 2008; Kim *et al*, 2012; Rauner *et al*, 2012). WNT signaling can be characterized as either the β-catenin-dependent canonical WNT pathway or the β-catenin-independent non-canonical WNT/planar cell polarity (PCP) pathway, which uses c-Jun N-terminal kinases (JNK), among others, for signal transduction. The canonical WNT signaling cascade is essential for normal cardiogenesis (Eisenberg & Eisenberg, 2006). WNT signaling in the adult heart, however, is silenced but reactivated after cardiac injury (Koval *et al*, 2011). There is an increasing body of evidence that reactivating the canonical WNT pathway negatively affects infarct healing with respect to cardiomyocyte death and cardiac fibrosis (Daskalopoulos *et al*, 2013). Yet the effects of modulating the non-canonical WNT pathway, in the context of myocardial healing, have barely been studied.

Modulating WNT signaling often occurs extracellularly via several WNT antagonists, which can be divided into two functional groups that both prevent ligand–receptor interactions. Members of the first group, Dickkopf (DKK), bind to WNT receptors and inhibit signal transduction, and the second group, secreted Frizzled-related protein (sFRP), bind directly to WNT proteins and block them from binding to their receptors (Kawano & Kypta, 2003; Clevers & Nusse, 2012). The WNT Inhibitory Factor 1 (WIF1) binds directly to WNT proteins (Hsieh *et al*, 1999). To date, WIF1 has not been studied in the context of either myocardial infarction or inflammation. Here we explain the local microenvironment's influence on the infarcted myocardium and monocyte activation. We found that WIF1 plays a key role during monocyte activation by controlling the inflammatory process after cardiac injury, thereby affecting cardiac function.

# Results

### The microenvironment orchestrates WNT signaling in inflammatory monocytes

In order to assess how traveling leukocytes adapt to their surroundings during inflammation, we isolated inflammatory Ly6C$^{hi}$ monocytes from mice 3 days after myocardial infarction. RNA-seq analyses revealed that MI results in a diverse transcriptional response in inflammatory monocytes sorted from the bone marrow (BM), blood, and heart. We observed differential expression of 1,482 genes in Ly6C$^{hi}$ monocytes from these locations (Fig 1A). Of note, we found similar expression patterns in Ly6C$^{hi}$ monocytes' transcriptome. A great proportion of the transcriptome was associated with WNT signaling (Fig 1B).

In addition, gene set enrichment analysis (GSEA) revealed highly and significantly increased genes bearing AP-1 and LEF1 binding

sites (Appendix Table S1) and differential expression of 39 WNT-associated genes (1C). LEF1 and AP-1 (Bengoa-Vergniory & Kypta, 2015) have been described to regulate WNT-induced gene expression of the canonical and non-canonical WNT signaling pathways, respectively. To further elucidate the regulation of different WNT signaling pathway branches in Ly6C$^{hi}$ monocytes from different regions, we further analyzed components of both the canonical and non-canonical WNT pathways. Comparing Ly6C$^{hi}$ monocytes isolated from infarcted myocardium and bone marrow showed increased numbers of canonical WNT inhibitors (i.e., members of the β-catenin destruction complex), whereas canonical WNT signaling mediators mostly decreased. By contrast, non-canonical WNT/PCP pathway mediators and target genes were upregulated in monocytes isolated from the infarcted heart compared to those from the bone marrow (Fig 1D and E).

To exclude changes in gene expression due to differences in cell processing, monocytes from the BM were exposed to the same isolation steps as those from the heart. Robust differences in key gene expression remained detectable between BM and heart monocytes—both isolated using digestion conditions—indicating that the observed changes are due to localization of the cells (Appendix Fig S1).

In addition, FACS analysis of ROR2, a key receptor for non-canonical Wnt signaling, was found to be upregulated in heart compared to bone marrow monocytes (Appendix Fig S2).

Phosphorylation of JNK is a crucial step in the activation of the WNT/PCP pathway. To further examine the role of the non-canonical WNT/PCP pathway, we evaluated phosphorylated JNK (pJNK) expression levels in post-MI murine heart whole-tissue lysates. Two days after MI, Western blot analysis showed increased pJNK levels (Fig 2A and B). *In vitro*, we found that hypoxic cardiomyocyte supernatant activates the non-canonical WNT pathway in macrophages through JNK and ATF2 phosphorylation. By contrast, the canonical WNT pathway was downregulated in the same macrophages, a result that corroborated the findings described above (Fig 2C and D and Appendix Fig S3). Another key mediator of JNK phosphorylation is TNFα (Jacobs *et al*, 1999). We could not detect significant differences in TNFα levels (data not shown) between supernatant of cardiomyocytes cultured under hypoxic and normoxic conditions indicating that TNFα might not be the driving force of JNK phosphorylation in our setup.

### WIF1 expression increases in hypoxic cardiomyocytes and is secreted in the border zone following MI

Since the local milieu seems to drive WNT regulation in accumulating monocytes, we evaluated several extracellular WNT antagonists in an *in vitro* MI model. In accordance with previous research, we found WNT to be either unaffected or attenuated after hypoxia (e.g., DKK1, SFrp5; Fig 3A). In contrast, the WNT antagonist WIF1 significantly increased in hypoxic cardiomyocytes (Fig 3A). However, hypoxia had no effect in regard to WIF1 expression levels in isolated fibroblasts (Fig 3B, left) and endothelial cells (Fig 3B, right), prompting us to focus on the interplay between cardiomyocytes and accumulating myeloid cells. *In vivo*, WIF1 was found to be elevated at early time points (days 1 and 3 post-MI) but return to baseline levels on day 7 after cardiac injury (Fig 3C and D). Immunofluorescence staining revealed WIF1 expression to be predominantly

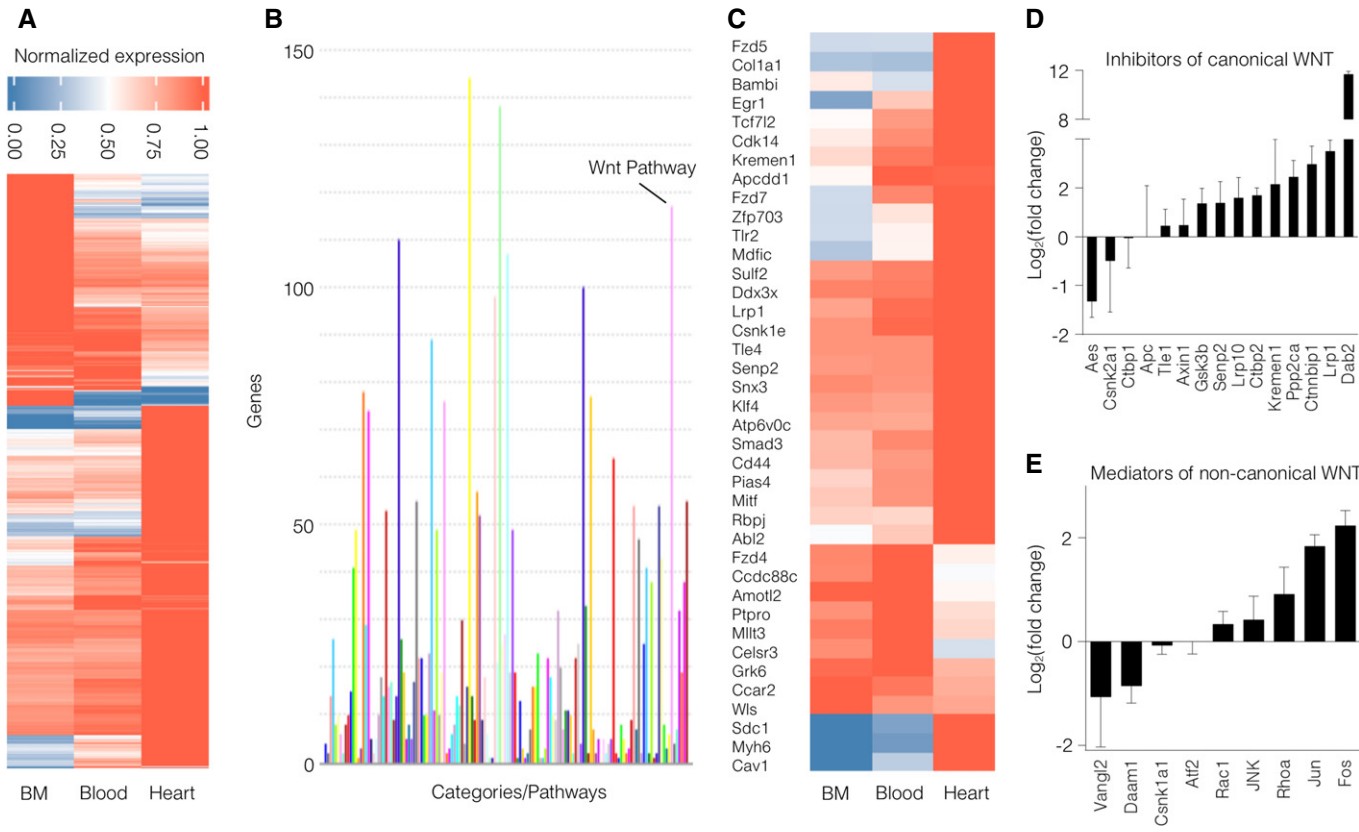

**Figure 1. Differential gene expression profiles in inflammatory monocytes sorted from the bone marrow, blood, and heart were found 3 days after MI.**

A   RNA-seq analyses revealed differential expression of 1,482 genes in monocytes sorted from different bodily regions.
B   PANTHER pathway analysis of genes found in the transcriptomes.
C   Differential gene expression of WNT-associated genes in monocytes.
D   Log$_2$(x-fold) of canonical WNT pathway inhibitors in Ly6C$^{hi}$ monocytes sorted from the heart compared to Ly6C$^{hi}$ monocytes in the bone marrow. Data are represented as mean ± SD (N = 3).
E   Log$_2$(x-fold) of non-canonical WNT/PCP pathway mediators in Ly6C$^{hi}$ monocytes sorted from the heart compared to Ly6C$^{hi}$ monocytes in the bone marrow. Data are represented as mean ± SD (N = 3).

located in the border zone of infarcted hearts (Fig 3E). In addition, WIF1 was detected in tissue sections from human patients with acute MI in contrast to patients who were not suffering from cardiac-related disease (Fig 3F) or ischemic cardiomyopathy (Appendix Fig S4).

**Absence of WIF1 results in impaired healing after MI**

The observed alterations in WIF1 encouraged us to further clarify WIF1's role in the post-MI healing process. We therefore induced MI in global WIF1 knockout animals and their WT littermates by permanent left anterior descending coronary artery (LAD) ligation. WIF1 knockout mice developed normally (Kansara *et al*, 2009) and showed no cardiac-specific phenotype under basal conditions, as evaluated by echocardiography (Appendix Fig S5). Upon LAD ligation however, we observed a dramatic increase in mortality in WIF1KO animals compared to their WT littermates within the first 72 h (Appendix Fig S6). In order to be able to conduct further analysis on infarct healing and remodeling, this prompted us to induce smaller infarct sizes than usual (as indicated by cardiac troponin T (cTnT) levels on day 1 after MI, Figs 4H and 5B). Four

weeks after MI, Masson trichrome staining revealed significantly increased infarct size in WIF1 KO animals, as compared to WT mice (Fig 4A and B). In addition, WIF1 KO animals showed a higher heart weight/body weight ratio than their WT littermates (Fig 4C). Echocardiographic analysis, moreover, revealed that WIF1 KO mice had developed severe cardiac dysfunction marked by significantly reduced fractional shortening (FS) and ejection fraction (EF), as compared to their WT littermates 4 weeks after MI (Fig 4D and E, see also Appendix Table S2 for a comparison of both groups).

Since ischemia triggers modulated WNT signaling in leukocytes, we evaluated the cellular inflammatory response in the presence and absence of WIF1. Flow cytometric analysis of mice hearts showed unaltered neutrophil numbers in WIF1 KO animals 1 day after induced MI, as compared to WT littermates (Fig 4F and G). Additionally, infarct size, as measured by cTnT levels in plasma samples, was unchanged in WIF1 KO mice and WT littermates 1 day after induced cardiac injury (Fig 4H). In contrast, the monocyte/macrophage response was critically altered 4 days after MI: the hearts of WIF1 KO animals had significantly more inflammatory

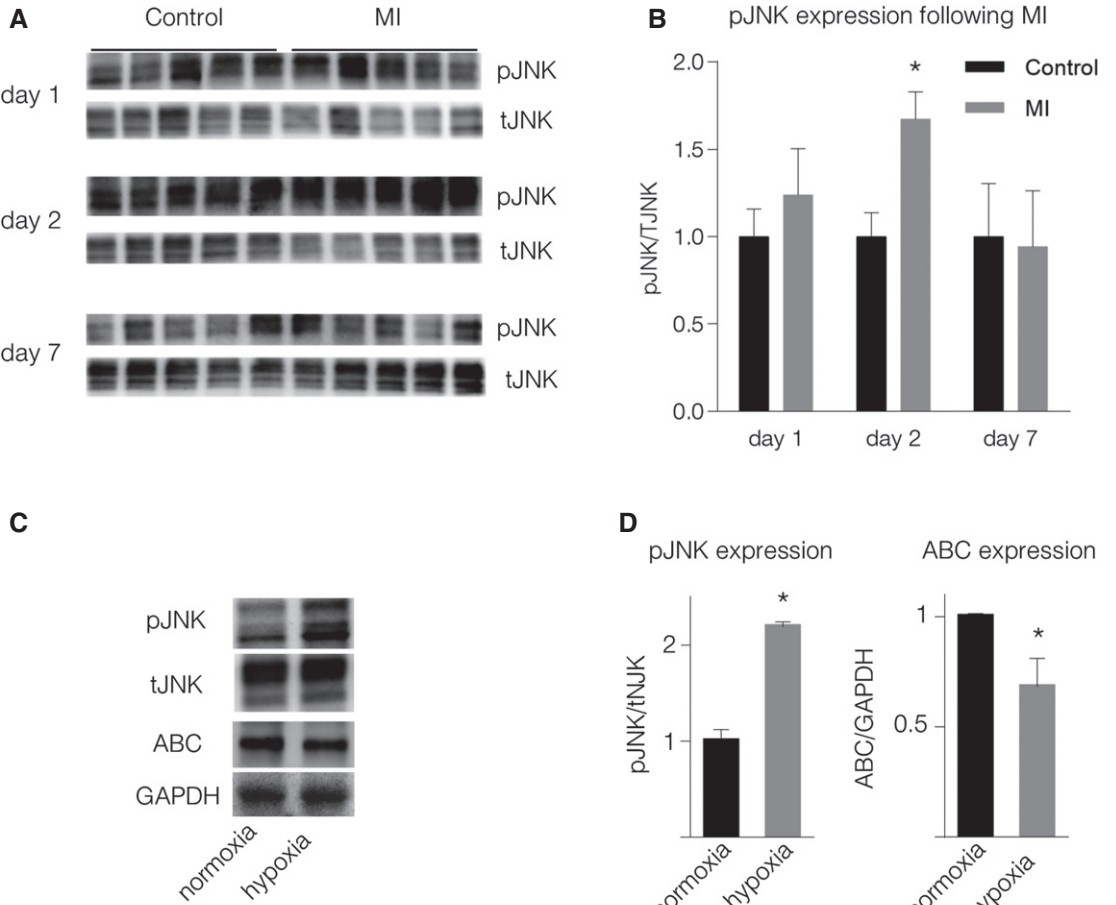

**Figure 2. Non-canonical WNT increases following MI.**

A      Representative Western blots of pJNK expression in the border zone of mouse hearts following MI.

B      Quantification of pJNK expression (mean ± SD, N = 5, *P = 8.6E-05, unpaired two-sided Student's t-test).

C, D   (C) Representative Western blots and (D) quantification of pJNK (*P = 0.0001) and active β-catenin (ABC) (*P = 0.001) expression in macrophages stimulated with supernatant of cardiomyocytes cultured under hypoxic conditions (unpaired two-sided Student's t-test, mean ± SD, N = 4).

Source data are available online for this figure.

Ly6C$^{hi}$ monocytes (Fig 4I and J; top left graph) and fewer reparative Ly6C$^{lo}$ macrophages than hearts from WT mice (Fig 4I and D; bottom left). Inflammatory monocyte levels in the blood and bone marrow did not change (Fig 4J; top right and bottom right).

Analysis of monocytes isolated on day 3 after LAD ligation from the hearts of WIF1 KO animals showed increased transcription levels of the downstream target of the AP-1/c-Jun transcription factor MMP13 (matrix metalloproteinase 13) and components of the non-canonical WNT signaling pathway ROR2, Rhoa, Rhou, Daam1, Dvl2 compared to monocytes isolated from wild-type animals (Appendix Fig S7). These findings may indicate that WIF1 impacts non-canonical WNT signaling in accumulating monocytes *in vivo*. To evaluate whether the WIF1 in myeloid cells itself impacts the immune response following MI, we performed bone marrow transfers from WIF1 KO mice into WT recipients. FACS analysis of hearts on day 4 after MI showed no significant differences between the groups' numbers of Ly6C$^{hi}$ monocytes and Ly6C$^{lo}$ macrophages (Appendix Fig S8). We therefore concluded that non-myeloid cells

rather than infiltrating immune cells are the main source of WIF1 in the heart after ischemic insult.

### Cardiomyocyte-specific WIF1 overexpression improves healing after MI

Because the absence of WIF1 impedes the post-MI healing process, we tested if WIF1 overexpression could enhance myocardial healing and improve heart function following cardiac injury. We therefore injected WT animals with either cardiotropic WIF1-AAV9 or a control LUC-AAV9 vector (under the control of a troponin T promoter) before inducing MI (see Fig 5A for timeline). A robust heart-specific WIF1 overexpression was achieved (see Appendix Fig S9). To evaluate whether cardiomyocyte-specific WIF1 overexpression alters infarct sizes at early time points, we measured cTnT levels in blood plasma 1 day after inducing myocardial infarction and found no differences between the groups (Fig 5B). Four weeks after MI, WIF1-overexpressing animals had reduced heart weight/body weight ratios (Fig 5C).

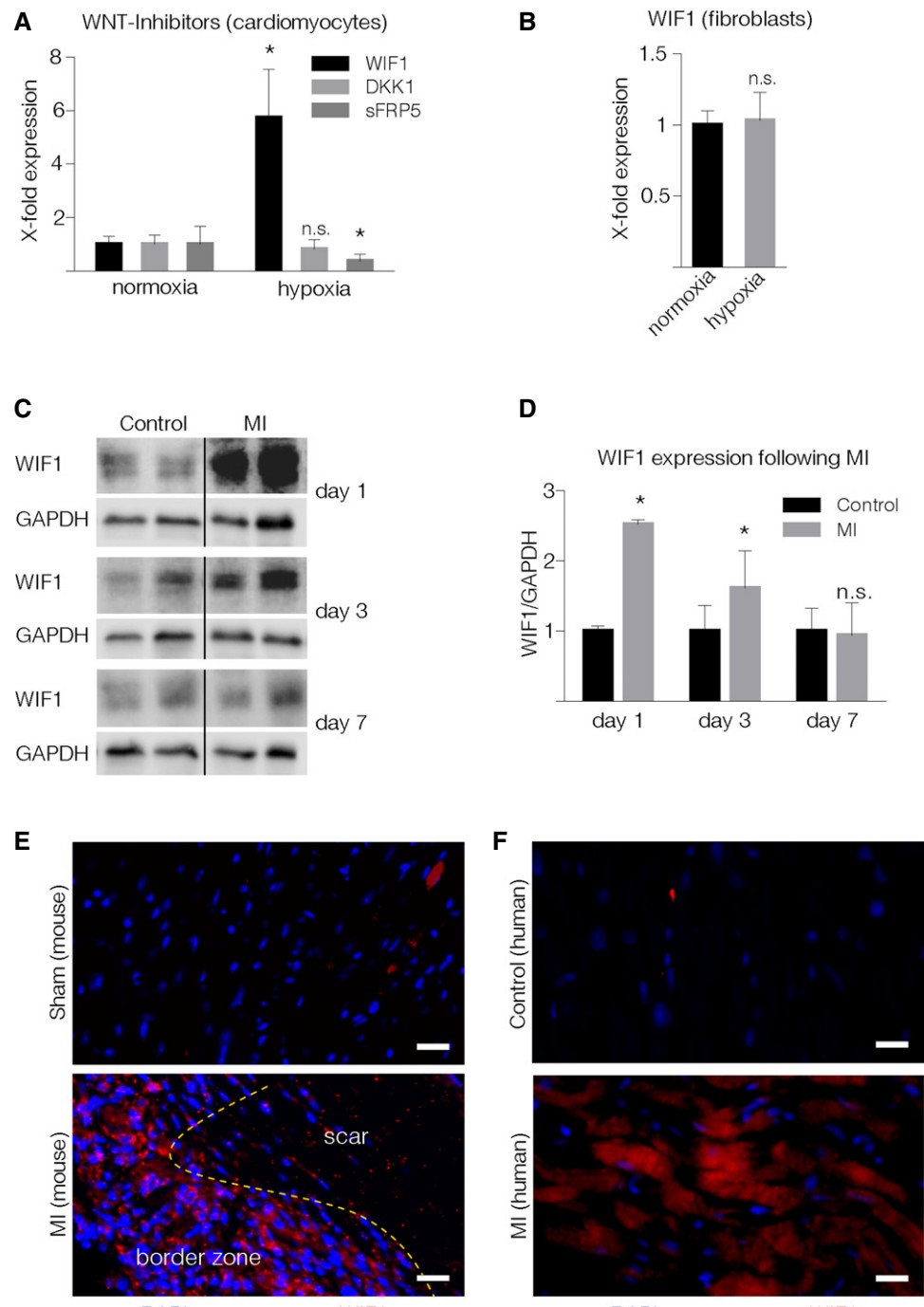

**Figure 3. WIF1 expression during MI.**

A    WNT inhibitor mRNA levels in isolated neonatal rat cardiomyocytes cultured under hypoxic conditions. Results from three independent experiments performed in triplicate (mean ± SD, WIF1: *$P$ = 0.011, sFRP5: *$P$ = 0.021; unpaired two-sided Student's $t$-test with Sidak correction).

B    WIF1 mRNA levels in isolated neonatal rat cardiac fibroblasts under hypoxic conditions. Results originate from three independent experiments performed in triplicate. Data are represented as mean ± SD.

C    Representative Western blots of WIF1 in hearts from sham-operated (control) or LAD-ligated (MI) mice.

D    Quantification of WIF1 protein expression in sham-operated and LAD-ligated animals (mean ± SD, $N$ = 4, Day 1: *$P$ = 0.0001, Day 3: *$P$ = 0.043; unpaired two-sided Student's $t$-test).

E, F    (E) Representative immunofluorescence staining of WIF1 in murine heart tissue sections from control (top), infarcted animals (middle), and (F) heart tissue sections from deceased human patients free from cardiovascular disease (top) or following acute MI (bottom) (red: WIF1, blue: DAPI). WIF1 expression in mice was visualized 4 days after LAD ligation. Scale bar = 25 μm. Yellow dotted line indicates the transition between scar and border zone (E).

Source data are available online for this figure.

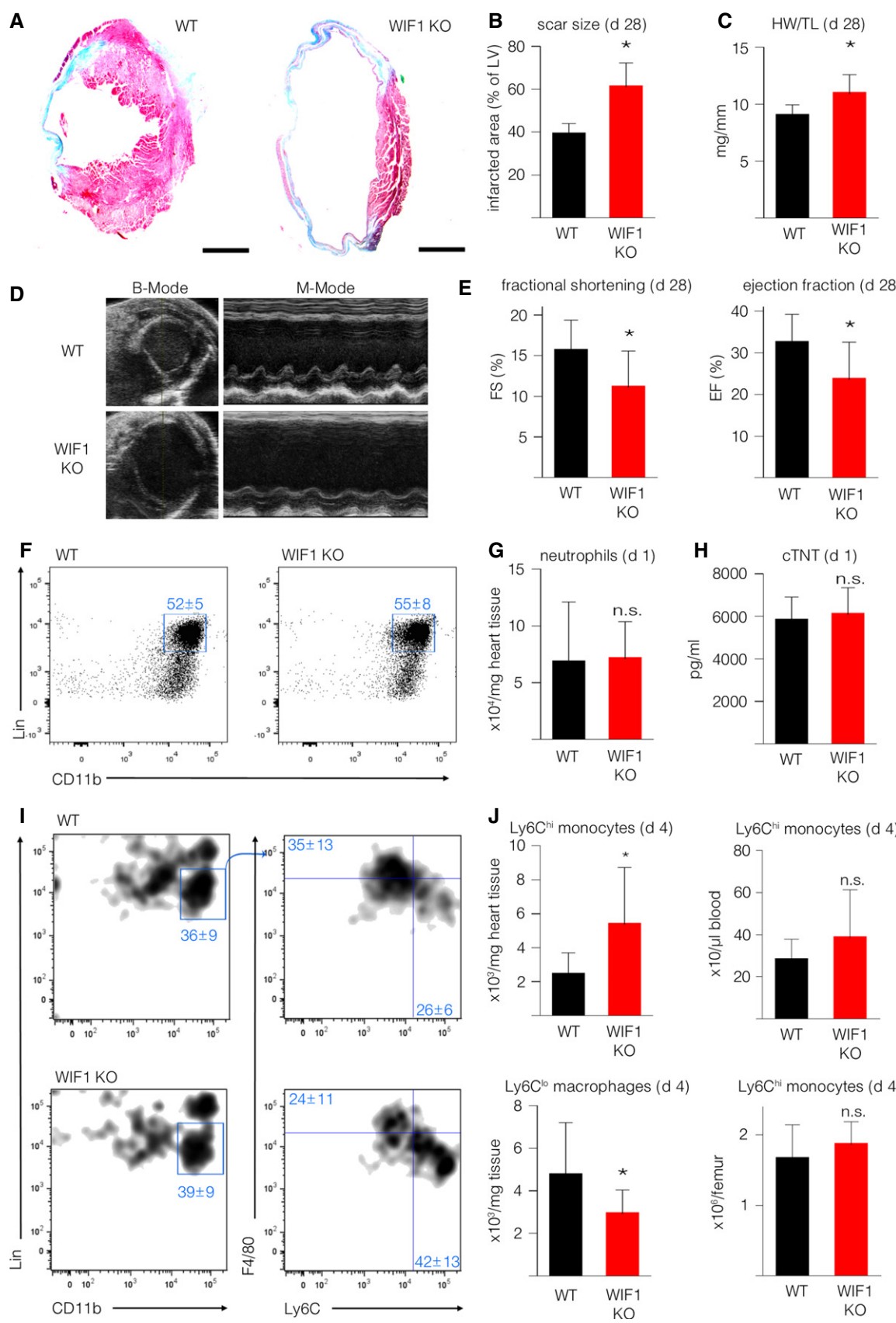

**Figure 4.**

**Figure 4.  Global WIF1 knockout worsens MI outcome.**

A   Representative Masson trichrome staining of WIF1KO (right) and their WT littermates (left) 4 weeks after induced MI (scale bar: 1,500 μm).
B   Quantification of relative scar size in WIF1KO and their WT littermates 4 weeks after MI (*P = 0.0001).
C   Heart weight/tibia length ratios 4 weeks after induced MI (*P = 0.0008).
D   Representative B-mode (left) and M-mode (right) echocardiographic images of WIF1KO (bottom) and their WT littermates (top) 4 weeks after MI.
E   Echocardiographic results from WIF1KO and their WT littermates 4 weeks after MI (N = 11, FS: *P = 0.0192, EF: *P = 0.0173).
F   Representative images of flow cytometric analysis of heart tissue cell suspension following MI in WIF1KO and WT mice gated on neutrophils 1 day post-MI.
G   Quantification of total neutrophils following MI (P = 0.91).
H   cTNT levels 1 day after LAD ligation (N = 11, P = 0.65).
I   Representative images of flow cytometric analysis of heart tissue cell suspension 4 days post-MI in WIF1KO (bottom) and WT (top) mice gated on monocytes/macrophages.
J   Quantification of total cell numbers per mg heart tissue of inflammatory (Ly6C$^{hi}$) monocytes (top left, *P = 0.0327); Ly6C$^{hi}$ monocytes per μl blood (top right, P = 0.1946); reparative (Ly6C$^{lo}$) macrophages per mg heart tissue (bottom left, *P ≤ 0.0439); and Ly6C$^{hi}$ monocytes per femur (bottom right, P = 0.3026).

Data information: Results are represented as mean ± SD and analyzed using unpaired two-sided Student's *t*-test (Sidak corrected). In panels (A–D, F, G, I, and J) N = 8 (WT) and N = 11 (WIF1-KO).

In addition, AAV9-WIF1 injected animals had significantly smaller scar sizes 4 weeks after induced MI (Fig 5D and E). WIF1 overexpression led to significantly improved FS and EF compared to control animals, a result that indicated ameliorated cardiac healing after MI (Fig 5F and G, see also Appendix Table S3).

In contrast to previous reports (Lu *et al*, 2013), we did not detect differences in cardiac function in WIF1-overexpressing versus control animals (Appendix Fig 10).

**Cardiomyocyte-specific WIF1 overexpression attenuates cardiac inflammation after MI**

Since we observed significantly different post-MI monocytic responses in WIF1 KO animals compared to WT mice, we next tested if cardiomyocyte-specific WIF1 overexpression impacts monocyte/macrophage numbers and the inflammatory response following MI. WIF1 overexpression led to reduced Ly6C$^{hi}$ monocyte numbers in the heart 4 days after MI (Fig 5H and I; top graph). The number of reparative Ly6C$^{lo}$ macrophages, however, did not significantly differ (Fig 5H and I; bottom graph).

**WIF1 ameliorates monocyte/macrophage inflammatory processes by inhibiting non-canonical WNT signaling**

Elegant binding analysis has previously shown that WIF1 interacts with the non-canonical WNT pathway's extracellular activators (Malinauskas *et al*, 2011). To evaluate whether cardiomyocyte-derived WIF1 affects myeloid cell activation, we induced WIF1 overexpression in isolated cardiomyocytes and exposed the cells to hypoxia. Supernatant of hypoxic cardiomyocytes activated JNK- and ATF2-phosphorylation in isolated monocytes/macrophages. This activation was significantly reduced after transfer of supernatant from hypoxic cells that overexpressed WIF1 (Fig 6A, C and D). In addition, monocytes/macrophages treated with supernatant of WIF1-overexpressing cardiomyocytes showed reduced transcription levels of pro-inflammatory cytokines IL-1β and IL-6, as compared to control cells (Fig 6B), thereby indicating that WIF1 reduces pro-inflammatory monocyte/macrophage activation by hypoxic cardiomyocytes.

# Discussion

Acute myocardial infarction causes a sterile, systemic inflammatory response. Ischemia and necrosis induce complement system activation, damage-associated molecular pattern molecules (DAMPs) release, and integrin activation on endothelial cells, thereby resulting in myeloid cell invasion at the injury site (Leuschner *et al*, 2012; Frangogiannis, 2014). Our findings suggest that upon accumulation, leukocytes are locally adjusted by WNT modulators actively secreted by cardiomyocytes. Previous reports have shown WNT signaling to be crucial following MI and ischemia reperfusion injury (Daskalopoulos *et al*, 2013; Bao *et al*, 2015; Nakamura *et al*, 2016); however, WNT signaling's precise role in regulating immune response—especially in the context of MI—is not yet understood. Here we show, for the first time, that the microenvironment is an important determinant of WNT signaling in the immune cell activation that leads to phenotypic changes in monocytes.

**Figure 5.  Cardiac-specific AAV-9-mediated WIF1 overexpression improves heart function following MI.**

A   Timeline of AAV-mediated WIF1 overexpression experiments.
B   cTNT levels 1 day after LAD ligation (N = 6, P = 0.19).
C   Heart weight/body weight ratios 4 weeks after induced MI (N = 6, *P = 0.0476).
D   Representative Masson trichrome staining of AAV-WIF1 (bottom) and AAV-LUC control animals (top) 4 weeks after induced MI (scale bar: 1,000 μm).
E   Quantification of relative scar size in AAV-WIF1 and AAV-LUC control animals 4 weeks after MI (N = 5, *P = 0.0264).
F   Representative B-mode (left) and M-mode (right) echocardiographic image of AAV9-LUC-injected control (top) and AAV9-WIF1-injected animals (bottom) 4 weeks after MI.
G   Echocardiographic results from AAV9-LUC-injected and AAV9-WIF1-injected mice 4 weeks after MI (N = 6, FS: *P = 0.0047, EF: *P = 0.038).
H   Representative images of flow cytometric analysis of heart tissue cell suspension following myocardial infarction of AAV-LUC-injected control (top) and AAV-WIF1 (bottom) mice.
I   Quantification of inflammatory (Ly6C$^{hi}$) monocytes (top, *P ≤ 0.04) and reparative (Ly6C$^{lo}$) macrophages (bottom, P = 0.58) per mg heart tissue (N = 12 per group).

Data information: Results are represented as mean ± SD and were analyzed by unpaired two-sided Student's *t*-test with Sidak correction.

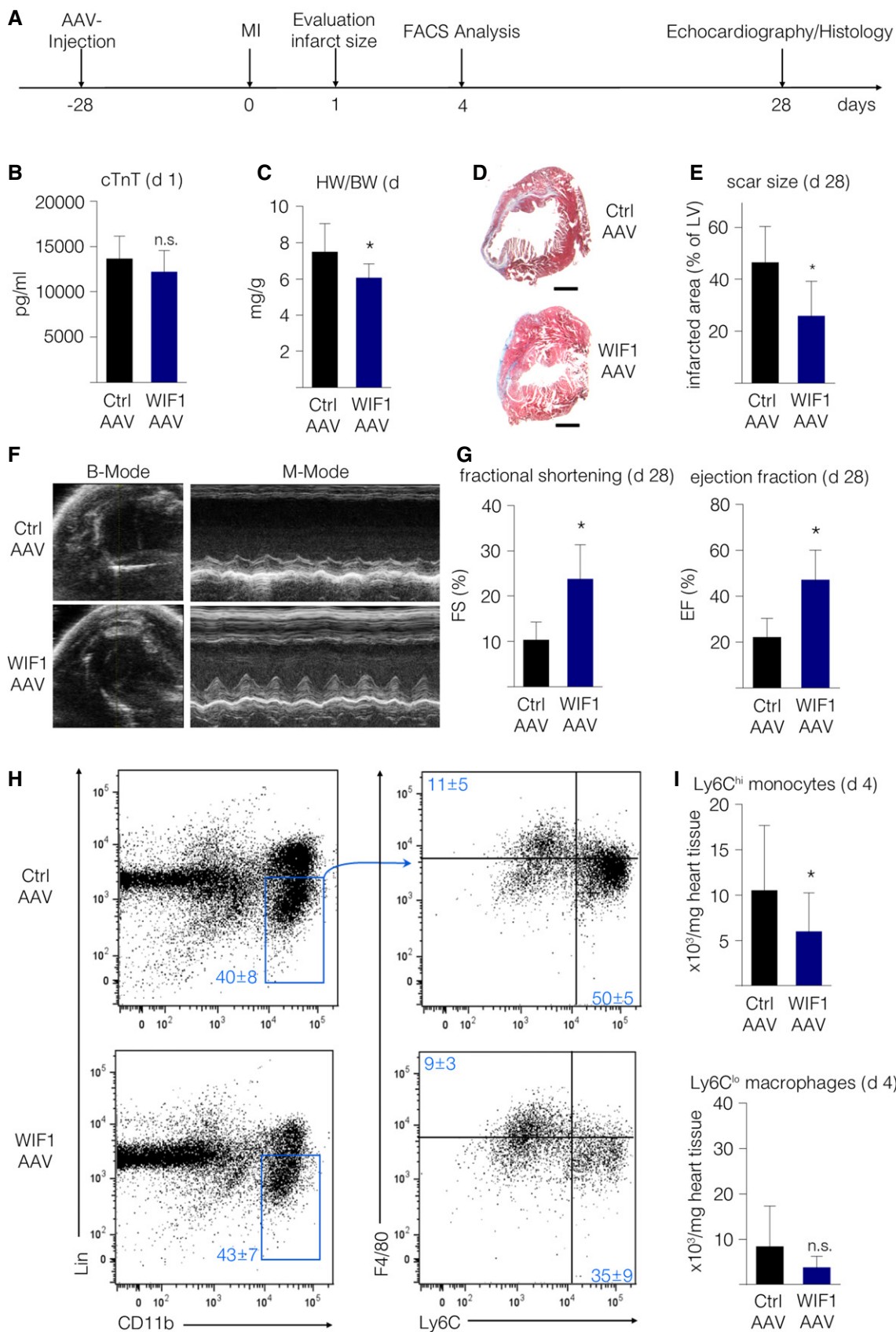

**Figure 5.**

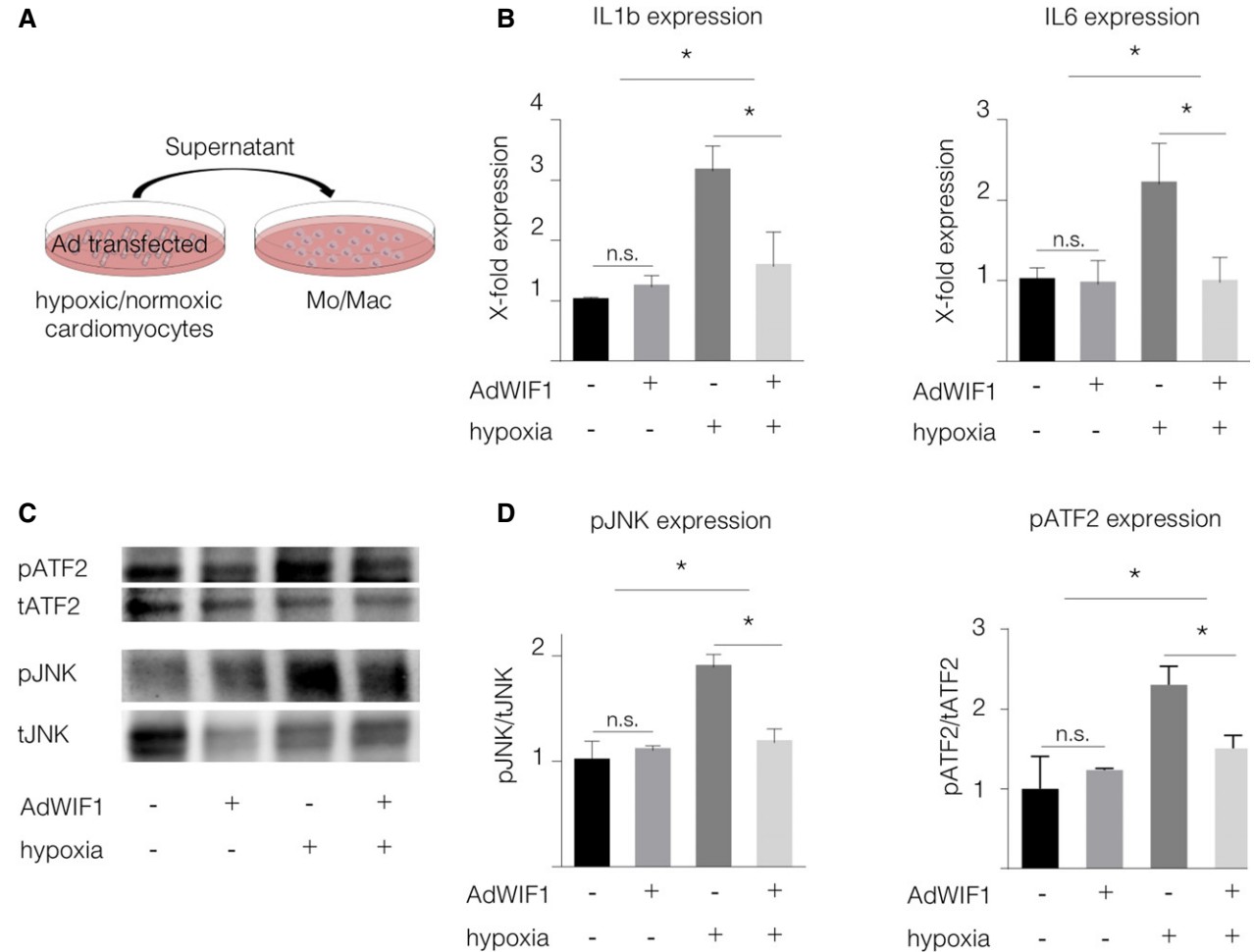

**Figure 6. WIF1 inhibits non-canonical WNT signaling.**

A    Scheme of *in vitro* experiments.
B    mRNA levels of inflammatory markers in macrophages stimulated with supernatant of control or WIF1 overexpressing cardiomyocytes cultured under hypoxic conditions (mean ± SD, N = 3, IL-1β: AdCtrl (normoxia) versus AdWIF1 (normoxia) P = 0.96 (n.s.), AdCtrl (normoxia) versus AdCtrl (hypoxia) *P = 0.000577, AdCtrl (hypoxia) versus AdWIF1 (hypoxia) *P = 0.004577. IL-6: AdCtrl (normoxia) versus AdWIF1 (normoxia) P = 0.99 (n.s.), AdCtrl (normoxia) versus AdCtrl (hypoxia) *P = 0.0156, AdCtrl (hypoxia) versus AdWIF1 (hypoxia) *P = 0.0141, one-way ANOVA with Holm-Sidak's multiple comparisons test).
C, D    (C) Representative Western blots of JNK and ATF2 expression in macrophages stimulated with supernatant of AdControl- or AdWIF1-transfected hypoxic cardiomyocytes and (D) Quantification of pJNK and pATF2 expression in macrophages stimulated with supernatant of AdWIF1-transfected hypoxic cardiomyocytes (mean ± SD, N = 4, JNK: AdCtrl (normoxia) versus AdWIF1 (normoxia) P = 0.99 (n.s.); AdCtrl (normoxia) versus AdCtrl (hypoxia) *P = 0.0001, AdCtrl (hypoxia) versus AdWIF1 (hypoxia) *P = 0.0001, ATF2: AdCtrl (normoxia) versus AdWIF1 (normoxia) P = 0.99 (n.s.); AdCtrl (normoxia) versus AdCtrl (hypoxia) *P = 0.0012, AdCtrl (hypoxia) versus AdWIF1 (hypoxia) *P = 0.0256, one-way ANOVA with Holm-Sidak's multiple comparisons test).

Source data are available online for this figure.

Our RNA-seq analysis shows that non-canonical WNT signaling is augmented in Ly6C[hi] monocytes isolated from the heart, but not the blood or bone marrow, in mice with myocardial infarction, thereby illustrating the microenvironment's interaction with recruited monocytes. We also found the canonical WNT pathway is suppressed in Ly6C[hi] monocytes at the site of inflammation. It was previously reported that canonical WNT signaling is activated in mononuclear bone marrow cells after MI (Assmus *et al*, 2012; Hermans *et al*, 2012). We found more components of the non-canonical WNT pathway and intracellular canonical WNT pathway inhibitors in monocytes isolated from the heart than in monocytes from the bone marrow. These data suggest that the non-canonical WNT

pathway may be activated in monocytes in the infarcted heart. Accordingly, non-canonical WNT signaling is activated in whole heart tissue during the first week after MI, a phase characterized by the presence of leukocytes in the heart. Despite these findings, one has to take into account that other cell types such as cardiomyocytes or fibroblasts may contribute to the observed activation of non-canonical WNT signaling in whole heart tissue. Changes in gene expression in monocytes might also occur due to the transition from bone marrow to blood and heart also beyond the impact of myocardial infarction. A clear limitation in this regard is that sorting inflammatory monocytes from sham-operated hearts for RNA-seq is not feasible.

In addition, our *in vitro* data suggest that troubled cardiomyocytes can activate non-canonical WNT signaling in monocytes directly, since monocyte/macrophage stimulation with hypoxic cardiomyocyte supernatant led to increased JNK and ATF2 phosphorylation and simultaneously decreased canonical WNT pathway. TNFα is known to mediate JNK phosphorylation and was found to be upregulated in heart tissue following acute MI (Jacobs *et al*, 1999). Although we were not able to detect differences in TNFα in the supernatant of cardiomyocytes cultured under normoxic and hypoxic conditions, we cannot exclude a contribution of TNFα signaling or other upstream pathways in non-canonical WNT/JNK signaling in monocytes/macrophages *in vivo*. Taken together, these findings support the idea that the non-canonical WNT signaling pathway is involved in inflammatory processes following myocardial infarction. Furthermore, these data show paracrine communication between cardiomyocytes and monocytes/macrophages has significant impact on both WNT signaling activity at the site of inflammation and the quality of recuperation after ischemic injury. This interaction appears to localize in the infarct border zone, which is both where most leukocytes accumulate in the injured myocardium and a crucial location for ongoing remodeling processes (Leuschner *et al*, 2010, 2011).

The inflammatory response following myocardial infarction is integral to adequate infarct healing. Neutrophils, Ly-6C$^{hi}$ monocytes, and Ly6C$^{lo}$ macrophages are responsible clearing necrotic tissue and developing a solid scar. However, an exaggerated inflammatory response aggravates infarct healing and may promote heart failure (Nahrendorf *et al*, 2007, 2010; Geissmann *et al*, 2010). Our research shows the extracellular WNT antagonist WIF1 seems to be a vital modulator of an adequate inflammatory process following cardiac injury. In addition, WIF1's presence in human heart tissue samples indicates its potential relevance in patients with myocardial infarction. In contrast to previous reports that suggest WNT antagonists decrease during MI (Bao *et al*, 2015; Nakamura *et al*, 2016), we found elevated WIF1 persisted during the inflammatory phase of myocardial healing. Although we cannot exclude the possibility that invading cells import additional WIF1, our *in vitro* studies show increased WIF1 in cardiomyocytes but not cardiac fibroblasts after hypoxia. Further, we observed *in vivo* that bone marrow transfer of WIF1 KO cells into WT mice did not alter the immune response. These results seem to indicate that cardiomyocytes are the crucial source of WIF1 after MI. The fact that cardiomyocyte-specific WIF1 overexpression was sufficient to modulate the monocyte response supports this hypothesis. Other groups have found WIF1 to be downregulated on an mRNA level post-MI. In contrast, we observed an increase in WIF1 protein levels which might be explained by different sampling of whole heart versus border zone tissue (Palevski *et al*, 2017).

Genetic WIF1 deficiency led to higher mortality, larger infarcts, reduced heart function, and increased inflammation. Augmented WNT signaling due to the absence of other WNT inhibitors or overexpression of WNT activators has been associated with increased cardiomyocyte death (van de Schans *et al*, 2008; Bergmann, 2010; Daskalopoulos *et al*, 2013). Upon adjustment to smaller infarct size induction however, our experiments with WIF1 KO and WT animals showed no difference in troponin T measurements from serum between the groups. We therefore hypothesized that reduced left ventricular function in WIF1 KO animals 4 weeks after myocardial

infarction resulted from an altered inflammatory response. Since transplanting WIF1 KO bone marrow cells into WT recipient mice had no effect on the inflammatory response following MI, we concluded that WIF1 acts not on cardiomyocytes but rather on the infiltrating immune cells in a paracrine manner. Yet WIF1 does not seem to alter the general inflammatory response. We detected no difference in neutrophil numbers between WIF1KO and WT animals, whereas monocyte and macrophage numbers were significantly altered. We therefore concluded that WIF1 activity is cell type specific and may fine-tune WNT signaling at the site of inflammation. The finding that isolated monocytes from WIF1 KO hearts on day 3 after MI showed increased transcription levels of MMP13—a downstream target of non-canonical WNT signaling—may indicate that the phenotype of an accumulating monocyte is altered in the presence of WIF1 *in vivo*. Given the increased numbers of pro-inflammatory Ly6C$^{hi}$ monocytes and reduced reparative Ly6C$^{lo}$ macrophages in WIF1 KO mice after MI, WIF1 may either accelerate the resolution of inflammation after MI or prevent the continuation of ongoing pro-inflammatory stimuli.

Finally, we investigated WIF1's protective potential in the context of myocardial infarction. Cardiomyocyte-specific WIF1 overexpression led to significantly fewer inflammatory monocytes in the heart after MI, while reparative macrophage numbers remained unaltered. This result challenges the hypothesis that WIF1 and non-canonical WNT pathway inhibition might be involved in the Ly6C$^{hi}$ monocytes' differentiation into Ly6C$^{lo}$ macrophages. Stimulating macrophages with supernatant of hypoxic cardiomyocytes that overexpress WIF1, however, led to not only reduced non-canonical WNT signaling activation but also reduced pro-inflammatory cytokine mRNA levels. These findings clearly indicate that WIF1 interferes with non-canonical WNT signaling in monocytes and macrophages and reduces pro-inflammatory activation (see schematic overview in Fig 7). AAV-mediated WIF1 overexpression supports myocardial healing and improves cardiac function after MI. While adeno-associated virus-based gene therapy has been approved for clinical practice (Flotte, 2013), the approach has risks and limitations. Alternative WIF1 modulation and administration should therefore be evaluated in the future.

We here describe the paracrine effect of cardiomyocyte-secreted WIF1 on monocytes. Other cell types such as cardiac progenitor cells (CPCs) might also be influenced by WIF1 as it has been reported that activation of WNT signaling can interfere the self-renewal of adult CPCs and blocks cardiac regeneration (Oikonomopoulos *et al*, 2011). WNT signaling has also been reported to play an important role in myofibroblasts formation and fibrosis in cardiac diseases (Duan *et al*, 2012; Blyszczuk *et al*, 2017). Blyszczuk *et al* and Duan *et al* found that inhibition of WNT signaling limits fibrosis and may be beneficial during the healing process following myocarditis and myocardial infarction.

Interestingly, Melgar-Lesmes and Edelman found that infiltrating monocytes colocalize with non-canonical WNT protein WNT5a following partial hepatectomy and may support vascular growth during liver regeneration (Melgar-Lesmes & Edelman, 2015). Inhibition of WNT signaling could therefore also lead to adverse effects regarding neovascularization following MI. These findings show the complexity of WNT signaling and the importance of understanding WNT signaling in a spatial-temporal manner.

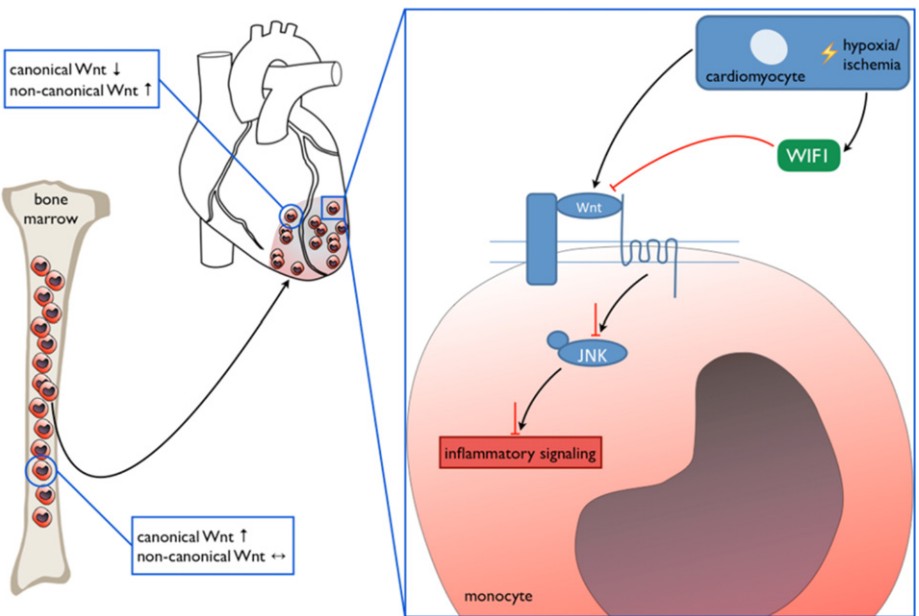

**Figure 7. Schematic overview of the identified mechanism of local monocyte activation.**
Schematic representation of proposed non-canonical WNT signaling activation in infiltrating monocytes by the cardiac microenvironment and inhibition by cardiomyocyte-derived WIF1 that limits inflammatory processes.

In summary, this study elucidates the interaction between the cardiac microenvironment and recruited monocytes. We demonstrate that the infarct's local surroundings activate non-canonical WNT signaling, which skews accumulating monocytes further into a pro-inflammatory state. WIF1 is a key player in restricting this local inflammatory monocyte response and thereby exhibiting cardioprotective properties. Therefore, WIF1 appears to be a promising target for future immunomodulatory approaches to improving healing after myocardial infarction.

# Materials and Methods

### Animals

WIF1 KO mice were provided by Dr. Igor B. Dawid (National Institutes of Health, Bethesda, USA). WIF1 KO were generated by inserting a tau-LacZ reporter cassette into the WIF1 coding sequence (Kansara *et al*, 2009). Animals were backcrossed for at least 6 generations and maintained on C57BL/6 background (Janvier, Saint-Berthevin, France). The procedure used male KO animals and their WT littermates, age 10–12 weeks. Cardiac-specific WIF1 over-expression in 6-week-old male C57BL/6 (Janvier, Saint-Berthevin, France) was reached by i.v. injection of $10^{12}$ AAV particles (for details on AAV production, see below). WIF1-overexpression was established for 4 weeks. Animals were housed under standard laboratory conditions with a 12-h light–dark cycle and access to water and food *ad libitum*. All experimental protocols were approved by the institutional review board of the University of Heidelberg, Germany, and the responsible government authority of Baden-Württemberg, Germany (project number 35-9185.81/G-27/14). A total of 145 animals were used for all experiments.

### Induction of myocardial infarction

Myocardial infarction was induced by permanent ligation of the left anterior descending coronary artery in male mice aged 10–12 weeks. In brief, anesthesia was induced with isoflurane (4%/800 ml $O_2$/min) and maintained by endotracheal ventilation (2–3%/800 ml $O_2$/min). Thoracotomy was performed in the fourth left intercostal space. The left ventricle was exposed, and the left coronary artery was permanently occluded. Chest and skin were closed, and anesthesia was terminated. Animals were extubated when breathing was restored. Initial myocardial injury was evaluated by measuring cardiac troponin T levels in plasma 24 h after induction of myocardial infarction. Sham-operated animals underwent the same procedure except for the occlusion of the LAD.

### Echocardiography

Echocardiography was performed on a Visualsonic Vevo 2100 4 weeks after induction of myocardial infarction. Mice were conscious during echocardiographic measurements. Ejection fraction (EF) and fractional shortening (FS) were determined based on M-mode measurements.

### AAV production

The AAV genome plasmid pdsTnT-mWIF1 was generated via amplification of the WIF1 sequence from murine cDNA with the primers WIF1 *Nhe*I fwd: 5′ TCAGTC*GCTAGC***GCCACC**ATGGCTCG GAGAAGAGC 3′ and WIF1 *Bsr*GI rev: 5′ TCAGTC*TGTACA*GGGTTC ACCAGATGTAATTGG 3′ (restriction sites *Nhe*I and *Bsr*GI underlined; KOZAK sequence bold) following subcloning to the vector pCR-(Zero)-Blunt. The correct sequence of the WIF1 cDNA was confirmed by sequencing following cloning to a self-complementary

AAV genome plasmid via restriction with *Nhe*I and *Bsr*GI. AAV vector production and purification was done using standard procedures. To purify the AAV, cell lysate Iodixanol step gradient centrifugation was used, as described elsewhere (Müller *et al*, 2006). AAV9 pseudotyped vectors were generated by co-transfection using the two plasmid system, which consists of helper plasmid pDP9rs and either genome plasmid pdsTnT-rluc or pdsTnT-mWIF1. pdsTnT-rluc contained a Renilla luciferase reporter gene under control of the human troponin T promoter and pdsTnT-mWIF1 contained the murine cDNA of WIF1.

## Flow cytometry and FACS sorting

Single-cell suspensions of infarcted hearts were obtained by mincing the tissue with fine scissors and digesting it with a solution containing collagenase I, collagenase XI, hyaluronidase (Sigma-Aldrich Chem GmbH, Taufkirchen, Germany), and DNase I (BD Biosciences, Heidelberg, Germany). $10 \times 10^6$ cells were stained for flow cytometric analyses. Inflammatory monocytes were identified as $Lin^-$(CD90;B220;CD49b; NK1.1;Ly6G;Ter119);$F4/80^-$;$CD11c^-$;$CD11b^+$; $Ly6C^{hi}$. Neutrophils were identified as $Lin^+$;$CD11b^+$;$F4/80^-$;$CD11c^-$;$Ly6C^{int}$. Flow cytometry was performed on FACS Verse (BD Biosciences, Heidelberg, Germany). Data were analyzed using FlowJo v10 (FLOWJO, LLC, Ashland, USA).

For FACS sorting, 10 animals underwent LAD ligation. Single-cell suspensions of heart and bone marrow were pooled. Inflammatory monocytes from infarcted hearts and bone marrow cell suspensions were sorted on FACS ARIAII (BD Bioscience, Heidelberg, Germany). RNA of sorted inflammatory monocytes was isolated using AllPrep DNA/RNA Micro Kit (Qiagen, Hilden, Germany). FACS sorting was conducted three times.

## RNA-seq analysis

RNA was quantified using a fragment analyzer (Advanced Analytical Technologies Inc., Heidelberg, Germany). 10 ng total RNA from sorted monocytes was used for library preparation. RNA-seq libraries were generated using TrueSeq RNA Access Library Prep Kit (Illumina, San Diego, USA). Libraries were then clustered at a concentration of 8 pmol, and sequencing was performed 2x100bl on a HiSeq2000 (Illumina, San Diego, USA) sequencer.

## Read processing and mapping

All reads were trimmed and quality clipped with Flexbar (Dodt *et al*, 2012). All remaining reads (> 18 bp in length) were mapped against the murine 45S rRNA precursor sequence (BK000964.3) to remove rRNA contaminant reads. We used the mouse genome sequence and annotation (EnsEMBL 79) together with the splice-aware STAR read aligner (release 2.5.1b) (Dobin *et al*, 2013) to assemble our conditions-specific target transcriptomes. Subsequent transcriptome analyses on differential gene and transcript abundance were carried out with the cufflinks package (Trapnell *et al*, 2012).

## Gene set enrichment analysis

Gene set enrichment analyses were carried out within the R framework (Dobin *et al*, 2013) using topGO (for GeneOntology) and EGSEA (for MSigDB) packages.

## Isolation of neonatal rat ventricular cardiomyocytes and hypoxia

Neonatal rat ventricular cardiomyocytes were isolated as previously described (Hagenmueller *et al*, 2010). After plating, NRVCMs were allowed to adhere overnight.

Cardiomyocytes, fibroblasts, and endothelial cells (HUVECs) were cultured under normoxic and hypoxic conditions (1.5% $O_2$) for either four or 24 h. Supernatant, RNA, and proteins were harvested. TNF$\alpha$ levels in supernatant were measured using TNF$\alpha$ ELISA Kit (R&D Systems, Wiesbaden, Germany). Supernatant was concentrated using Amicon 10K centrifugal filters (Millipore, Darmstadt, Germany).

## AdWIF1 transfection of cardiomyocytes

Premade WIF1 and control adenovirus was purchased from Applied Biological Materials Inc. (Richmond, Canada). Adenovirus was amplified according to the manufacturer's instructions. Cardiomyocytes were transfected according to manufacturer's instructions. WIF1 overexpression was allowed to establish for 48 h. Transfected cardiomyocytes were then cultured under hypoxic conditions as previously described.

## Isolation and stimulation of PBMC-derived macrophages

Adult rats were euthanized using a high dose of pentobarbital. Blood was collected by cardiac puncture into EDTA-coated sample tubes. Mononuclear cells were separated using Histopaque 1083 (Sigma-Aldrich Chemie GmbH, Taufkirchen, Germany) according to manufacturer's instructions. Monocytes/macrophages were allowed to adhere to the cell culture dish for 2 h. Non-adherent mononuclear cells were discarded. Macrophages were stimulated with supernatant of hypoxic cardiomyocytes for 4 h.

## Western blot analysis

Hearts of MI- and sham-operated animals were extracted. We then separated the border zone from the infarcted area and used the border zone for lysate preparation. Protein lysates of border zone heart tissue and primary cells were prepared using RIPA buffer supplemented with phosphatase/proteinase inhibitors (Cell Signaling Technology, Danvers, USA). Protein concentrations were measured using BCA assay (Thermo Fischer Scientific, Waltham, USA). Equal amounts of protein were separated onto 4–15% SDS–PAGE gradient gels (Bio-Rad Laboratories GmbH, München, Germany) and transferred to PVDF membranes (Merck Chemicals GmbH, Darmstadt, Germany). The following primary and HPR-conjugated secondary antibodies were used: monoclonal mouse anti-active-β-catenin (anti ABC), clone 8E7 (Merck Millipore, Darmstadt, Germany); monoclonal rabbit anti-JNK1+JNK2+JNK3 [EP1597Y] (phosphor Y223+Y185+Y185) (Abcam, Cambridge, UK); monoclonal rabbit anti-JNK1+JNK2+JNK3 antibody [EPR16797-22] (Abcam, Cambridge, UK); monoclonal rabbit anti-WIF1 [EPR9385] (Abcam, Cambridge, UK); monoclonal rabbit anti-GAPDH [EPR16891] (Abcam, Cambridge, UK); anti-ATF2 (phosphor T51 + T69) (Abcam, Cambridge, UK); polyclonal rabbit anti-ATF2 antibody (Abcam, Cambridge UK); HRP-conjugated goat anti-rabbit IgG (Abcam, Cambridge, UK). Proteins were visualized using Amersham ECL Western blotting detection reagents (GE Healthcare Europe GmbH, Freiburg, Germany). Images were captured using Peqlab

Fusion FX (Peqlab Inc., Erlangen, Germany). Protein expression was analyzed using ImageJ. Phosphorylated JNK was normalized to total JNK (tJNK). Phosphorylated ATF2 was normalized to total ATF2. Other proteins were normalized to GAPDH.

### Quantitative real-time PCR

Total RNA was isolated from primary cells using Trizol reagent (Thermo Fischer Scientific, Waltham, USA) following manufacturer's instructions. RNA was reverse transcribed using Revert Aid First Strand cDNA Synthesis Kit (Thermo Fisher Scientific, Waltham, USA). RT-qPCR was performed using SYBR Green (Bio-Rad Laboratories GmbH, München, Germany) according to manufacturer's instructions. Gene expression was normalized to HPRT. The following primers were used: WIF1: TTCTTTAAAACATGTCAACAAGCTG (fwd), ACAAAAGCCTCCATTTCGAC (rev); HPRT: GTCAACGGGG GACATAAAAG (fwd), TGCATTGTTTTACCAGTGTCAA (rev); IL-1β: TCGTGCTGTCTGACCCATGT (fwd), ACAAAGCTCATGGAGAATAC CACT (rev); IL-6: AACTCCATCTGCCCTTCAGGAACA (fwd), AAGG CAGTGGCTGTCAACAACATC (rev); LRP1: GGACCACCATCGTGGAA (fwd), TCCCAGCCACGGTGATAG (rev); DAB2: TCTCAGCCTG CATCTTCTGA (fwd), TTTGCTCATCTGGATAGTCATCAT (rev); FOS: GGGACAGCCTTTCCTACTACC (fwd), AGATCTGCGCAAAAGTCCTG (rev); JUN: CCAGAAGATGGTGTGGTGTTT (fwd), CTGACCCTCTCCC CTTGC (rev); ROR2: TCATCAGCCAGCACAAACA (fwd), GTGG CCTTTGTAGACCTTGC (rev); MMP13: CAGTCTCCGAGGAGAAACT ATGAT (fwd), GGACTTTGTCAAAAAGAGCTCAG (rev); DAAM1: GTTGGCCCGAGTTCTACATT (fwd), TCTAAAGCCAGCAGAGATTT CC (rev); RhoA: ACCTGTGTGTTTTCAGCACCT (fwd), CCATCACC AACAATCACCAG (rev); RhoU: ACGGCCTTCGACAACTTCT (fwd), ACTCATCCTGTCCTGCAGTGT (rev); DVL2: TATGTCTTCGGGGA CCTCAG (fwd), GGACCGTCATTGTCATTCA (rev).

### Immunofluorescence WIF1 staining

Paraffin-embedded heart tissue sections from patients who died of acute myocardial infarction, with cardiomyopathy, and from patients who died from a non-cardiac-related disease were provided by the tissue bank of the National Center for Tumor Diseases (NCT, Heidelberg, Germany) in accordance with its regulations and the approval of Heidelberg University's ethics committee. Paraffin-embedded mouse and human heart tissue sections were deparaffinized using xylol and rehydrated in decreasing ethanol concentrations. Antigen retrieval was performed using citrate buffer for 15 min at 90°C. Sections were stained with polyclonal goat anti-rabbit WIF1 (Abcam, Cambridge, UK). WIF1 antibodies were fluorescently labeled using goat anti-rabbit Alexa Fluor 694 (Abcam, Cambridge, UK). Images were acquired on an Axio Observer (Carl Zeiss Microscopy GmbH, Jena, Germany). For further details on histological sampling and analysis, see Appendix Supplementary Methods.

### Statistics

Statistical analyses were performed using GraphPad Prism 6 (GraphPad Software Inc., La Jolla, USA). Data are represented as mean ± SD. Differences between two groups were analyzed by Student's *t*-test with correction for multiple testing using Dunn–Sidak method. Differences between more than two groups were

---

## The paper explained

### Problem

Myocardial infarction (MI) is a leading cause of death worldwide. The inflammatory response following cardiac injury greatly impacts healing and recovery of left ventricular function after MI. Monocytes are key players in this process.

### Results

We here investigated differences in local versus systemic activation of monocytes following MI. Of note, comparison of monocytes from the site of production (bone marrow) and the site of inflammation (heart) showed a strong differential regulation of WNT signaling pathways. Further analysis revealed that non-canonical WNT signaling in monocytes is activated by the cardiac microenvironment and may skew the accumulating leukocyte into a pro-inflammatory state. This mechanism of local activation is controlled by WNT Inhibitory Factor 1 (WIF1), a protein secreted by cardiomyocytes. Adeno-associated virus (AAV)-mediated overexpression of WIF1 resulted in reduced cardiac inflammation and improved cardiac function after MI.

### Impact

The obtained results illustrate the potential of WNT modulation by WIF1 as a promising target to prevent heart failure following ischemic injury.

---

analyzed by one-way ANOVA followed by Sidak *post hoc* analysis. *P* < 0.05 was considered to be statistically significant.

### Study approval

Animal studies were approved by the regulatory authorities (Regierungspräsidium Karlsruhe of the state of Baden-Württemberg/ Germany, 35-9185.81/G27/14). Human tissue samples were provided by the tissue bank of the National Center for Tumor Diseases (NCT, Heidelberg, Germany) in accordance with the regulations of the tissue bank and the approval of the ethics committee of Heidelberg University (Project number 1974 and project number 2141).

**Expanded View** for this article is available online.

### Acknowledgements

The authors thank Dr. Igor B Dawid (National Institute of Health, Bethesda, USA) and Prof. Dr. Heidi Hahn (Institute of Human Genetics, University of Göttingen, Göttingen, Germany) for providing WIF1 knockout mice; the Benjamin Meder group (Department of Medicine III, University of Heidelberg, Heidelberg, Germany) for RNA sequencing and Monika Langlotz (FACS Core Facility, Zentrum für Molekulare Biologie der Universität Heidelberg, Heidelblberg, Germany) for monocyte cell sorting.

### Author contributions

ISM, MB, MN, HAK, SEH, and FLe conceived and designed the study. ISM and MZ performed LAD ligation surgery and *ex vivo* analyses. ISM and FLe performed echocardiography. ISM and FLe performed and analyzed FACS analyses. AJ and OJM designed and produced AAV9-vectors. JH performed RNA sequencing analysis. CD performed read processing and gene enrichment analysis. ISM and SW performed *in vitro* experiments. ISM and FLa performed immunofluorescence stainings.

## Conflict of interest

The authors declare that they have no conflict of interest.

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
