## [Review Process File · EMBO Molecular Medicine]

The cardiac microenvironment uses non-canonical WNT signaling to activate monocytes after myocardial infarction

I Sören Meyer, Andreas Jungmann, Christoph Dieterich, Min Zhang, Felix Lasitschka, Susann Werkmeister, Jan Haas, Oliver J Müller, Michael Boutros, Matthias Nahrendorf, Hugo A Katus, Stefan E Hardt, Florian Leuschner

Corresponding author: Florian Leuschner, University of Heidelberg

Review timeline:

Submission date:	10 January 2017
Editorial Decision:	07 February 2017
Revision received:	02 May 2017
Editorial Decision:	02 June 2017
Revision received:	07 June 2017
Accepted:	20 June 2017

Transaction Report:

Editor: Roberto Buccione

1st Editorial Decision

07 February 2017

Thank you for the submission of your manuscript to EMBO Molecular Medicine. We have now heard back from the Reviewers whom we asked to evaluate your manuscript.

As you will see, the detailed and thoughtful evaluations are quite complementary and, in my opinion draw a rather clear picture. On one hand, the reviewers all agree that the study is interesting, has merits and has potential clinical relevance and translational implications. On the other hand, they also point to significant limitations encompassing various key issues. These include the unresolved/non-discussed conflict with existing knowledge, lack of crucial controls, insufficient mechanistic insight and therefore experimental support for the main conclusions, poor data presentation and manuscript organization in some instances. The reviewers also list other items for consideration.

After reviewer cross-commenting and further discussion, it was agreed that you should be allowed to submit a substantially revised manuscript, with the understanding that the reviewers' concerns must be addressed in full with additional experimental data where appropriate and that acceptance of the manuscript will entail a second round of review.

It is important that you consider that it is EMBO Molecular Medicine policy to allow a single round of revision only and that, therefore, acceptance or rejection of the manuscript will depend on the completeness of your responses included in the next, final version of the manuscript.

As you know, EMBO Molecular Medicine has a "scooping protection" policy, whereby similar

findings that are published by others during review or revision are not a criterion for rejection. However, I do ask you to get in touch with us after three months if you have not completed your revision, to update us on the status. Please also contact us as soon as possible if similar work is published elsewhere.

Finally, please note that EMBO Molecular Medicine now requires a complete author checklist (<http://embomolmed.embopress.org/authorguide#editorial3>) to be submitted with all revised manuscripts. Provision of the author checklist is mandatory at revision stage; The checklist is designed to enhance and standardize reporting of key information in research papers and to support reanalysis and repetition of experiments by the community. The list covers key information for figure panels and captions and focuses on statistics, the reporting of reagents, animal models and human subject-derived data, as well as guidance to optimise data accessibility.

Please note that we now mandate that all corresponding authors list an ORCID digital identifier. You may do so through our web platform upon submission and the procedure takes <90 seconds to complete. We also encourage co-authors to supply an ORCID identifier, which will be linked to their name for unambiguous name identification.

I look forward to seeing a revised form of your manuscript in due time.

***** Reviewer's comments *****

Referee #1 (Remarks):

This manuscript explores the impact of the cardiac local microenvironment on the infarcted myocardium and monocyte activation. Specifically, the authors found that the Wnt inhibitory factor 1, WIF1, plays a key role during monocyte activation by controlling the inflammatory process after myocardial infarction (MI). RNAseq analysis shows that non-canonical WNT signaling is increased in Ly6Chi monocytes isolated from the heart, but not from the blood or bone marrow, in mice following MI, reflecting the microenvironment's interaction with recruited monocytes. In vitro analysis revealed a strong upregulation of WIF1 in isolated neonatal cardiomyocytes - but not fibroblasts or endothelial cells - upon hypoxia. In vivo, WIF1 was found to be elevated at early time points after MI (days 1 and 3 post-MI). The important role of WIF1 is underscored by the work done of WIF1 KO mice. Compared to WT, WIF1 KO mice showed impaired cardiac function marked by increased scar size and reduced ejection fraction four weeks after MI and the heart of WIF1 KO contained significantly more inflammatory monocytes than those of WT mice. Further exploring the modulatory role of WIF1, the authors induced AAV-mediated cardiomyocyte-specific WIF1 overexpression, which attenuated the monocyte response and improved cardiac function after MI, as compared to control-AAV-treated animals. Finally, WIF1 overexpression in isolated neonatal cardiomyocytes limited the activation of non-canonical WNT signaling and led to reduced IL1 β and IL6 expression in monocytes/macrophages. In light of their results, the authors suggest WIF1 is a key player in restricting the local inflammatory monocyte response, and thereby exhibits cardioprotective properties.

Overall, this study reveals interesting findings with potentially important clinical relevance. While the data are sound, they are not always clearly presented and they suffer from significant weaknesses that need to be addressed along the following points:

Major concerns:

1- In contrast to previous reports suggesting that WNT antagonists (frizzled-related protein 5 and DKK3) decreased following MI, the authors found elevated WIF1 during the inflammatory phase of myocardial healing (Figure 3). However, a recent study showed that 4 days post-MI a downregulation (2.2-fold) of WIF1 was observed in the heart (Palevski et al, *J Am Heart Assoc.* 2017;6:e004387. DOI: 10.1161/JAHA.116.004387). The same group also found a downregulation of GSK3 β , while in the current manuscript an up-regulation of GSK3 β was observed early following MI. The authors should clarify this issue.

2-To validate the role of the non-canonical WNT/PCP pathway, the authors evaluated phosphorylated JNK (pJNK) expression levels in post-MI murine heart whole-tissue lysates and they found increased pJNK levels until day 7 post-MI as compared to sham-operated animals. In figure 2 A and B, the data are normalized to actin expression. However, the potential changes are

highly dependent on the extent of the induced infarct. The data in general should be first normalized to the cardiac Troponin T (cTnT) levels monitored on day 1 after MI. This holds also for figures 3-5. 3- In Figure 3F, an immunostaining of WIF1 is provided with tissue sections from human patients with acute MI. A control tissue section of WIF1 should be provided from an individual not suffering from MI or a related cardiac disease.

4- In Figure 6, supernatant of hypoxic cardiomyocytes activated JNK phosphorylation in isolated monocytes/macrophages. This activation was significantly reduced after transfer of supernatant from hypoxic cells that overexpressed WIF1. To validate the impact of WIF1 on non-canonical WNT signaling in accumulating monocytes, did the authors check the MMP13 levels in isolated monocytes/macrophages?

5- The potential paracrine impact of ischemic cardiomyocytes via WIF1 on local monocytes, should be also discussed in light of the possible effects on the niche of cardiac progenitor cell self-renewal. Minor concerns:

1- The supplementary figures should be re-numbered, according to the order of presentation of the data in the results section. For example, at the beginning of the results, supplementary fig. 6 is presented, why it is not numbered as Suppl. Fig 1.

2- While described in the text, no supplementary tables (1 and 2) were provided in the files.

3- The title of the legend of figure 3 should be: "WIF1 expression during MI: and not "WIF1 protein expression during MI", because both mRNA and protein data are provided.

4- The data on MMP13 in monocytes (p.9) are not quoting the supplementary fig.8 that describes them.

Referee #2 (Remarks):

This is a well-written and clearly structured manuscript describing the identification of non-canonical WNT pathway modulators in the ischemic heart with a role in shaping the inflammatory and reparative response of the infiltrating monocytes. It argues for a specific role of the cardiac microenvironment (specifically the injured cardiomyocytes themselves) in controlling monocyte activity via the WNT pathway. Specifically, the non-canonical WNT modulator WIF1 was uncovered as a key player in this context. Notably, the causal role of WIF1 was shown by global KO and AAV-based cardiac overexpression. As controlling the inflammatory response in the first days after MI is viewed as a promising avenue to prevent adverse remodeling and heart function insufficiencies following MI, and although the current study is not (yet) fully convincing at this point, the data offer promise for novel anti-inflammatory strategies for ischemic cardiac injury. The WNT pathway is a molecular axis that had not been considered in this context. The paper therefore offers a good degree of novelty. However, some of the conclusions (in particular those related to JNK) are only partly supported by the data shown and some of the figures are of poor/insufficient quality to justify the conclusions. Overall, the molecular link between non-canonical WNT signaling and the JNK pathway is not yet convincingly shown.

The following major criticism should be addressed:

1. Where the RNAseq hits confirmed by qPCR?
2. It is surprising that typical non-canonical WNT regulators like Sfrp5 or WNT5a are not showing up in the transcriptome analysis. How can this be explained?
3. What ROR2, a typical non-canonical WNT receptor?
4. The meaning of Suppl. Fig. S6 is not really clear.
5. A main weakness of the paper is that other upstream pathways (receptors) that can lead to JNK activation in the ischemic heart are not considered at all (TNF, CD74 signaling etc.). These should be ruled out by specific blocking experiments to underpin the role of the WNT pathway.
6. Also, while generally well-written with a nice flow for the reader to follow, the transition from Figure 1D/E to JNK as a main WNT-driven culprit pathway is hard to follow for the reader. Actually, the entire JNK part is a bit confusing. The expression of total JNK is discussed in Figure 1E, but Figure 2 then somehow switches to pJNK. Total JNK levels usually are rather stable, but seem to be upregulated as part of non-canonical WNT activation in the heart.
7. Figure 2 is unconvincing. First of all, it is strange that only one JNK band (which one: 46 or 54 kD?, full blot for Figure 2A?) shows up. In contrast, in Figure 6A, 2 JNK bands can be seen. Molecular weight markers are missing. Secondly, what about total JNK. If pJNK as a measure of short-term JNK activation is measured, total JNK levels must be used for standardization rather than

beta-actin. Also, the quality of the blots (and bands is rather poor).

8. Along the same lines, other targets than MMP13 should be measured to establish the link between WIF1 and non-canonical WNT signaling in the accumulating monocytes.

Minor:

1. Figure 3E: size bars are missing.
2. It is unclear why a pie diagram was used in Figure S7? How many mice do these percentage numbers refer to? I cannot find this number?
3. The BMT experiment from WIF1 KO mice is not well explained. What else was measured? Was there an impact on infarct size and/or heart function? What are the conclusions?

Referee #3 (Remarks):

The authors provide an interesting view on the role of Wnt signalling during myocardial infarction. Their differentiation between the canonical and non-canonical signalling can be very relevant for the interpretation of previous and ongoing studies. The manuscript is well written. The authors provide data in support of their conclusions. Especially the evidence regarding cardiomyocytes as main source of WIF1 is supported by clear in vitro and in vivo data. However, there are some critical issues that need to be addressed before a final conclusion can be drawn.

General comments

- RNA sequencing provides an unbiased approach to identify transcriptional changes. However, the method highly depends on sample preparation and input, and the authors describe how monocytes have been sorted by FACS. Therefore, it is surprising to find MYH6, a highly specific marker for cardiomyocytes, among the enriched genes in heart-derived monocytes (figure 1C). The same holds to a lesser extent for Colla1, which is also highly expressed by myocytes and fibroblasts. It would be good if the authors reflect on this and provide additional evidence that the sorting was performed properly.
- In line with the previous comment: it can be argued that there are many other factors, besides the myocardial infarction, that underlie the changes in gene expression when comparing monocytes from bone marrow, blood, and heart. I am aware that this is probably the only feasible approach, but suggest that the authors discuss this limitation.
- The authors ascribe the changes in cardiac pJNK and pATF2 to the infiltrating monocyte fraction (figure 2). JNK and ATF2 expression is however not restricted to monocytes, and it is unlikely that the observed effect can only be ascribed to infiltrating monocytes.
- The conclusion that effects of WIF1 inhibition result from changes in MMP13 expression is quite strong, and it would be good to provide some additional evidence for changes in ECM remodelling (figure 1C, for example, actually shows upregulation of Colla1 expression, which would speak for the contrary).
- The authors fail to put their observations in a broader perspective. For example: WIF1 has been shown to play a role in early angiogenesis (Melgar-Lesmes et al. PMID: 26022689), and WNT-signalling regulates myofibroblast recruitment (Blyszczuk et al. PMID: 27099262). The discussion would greatly benefit from a broader review of (contradicting) evidence.

Figures

- Figure 1: provide an overview of the gating strategy, eventually as supplemental figure. See comments above.
 - Figure 2A,C and Figure 6A: It can be argued that pJNK and pATF2 represent the active and relevant read-out, but it would be very informing to show western blots of total JNK and ATF2 levels.
 - Figure 3A-C: Is there any information regarding "absolute" levels of expression of WIF1 in the cell types? The conclusions are affected if its expression in cardiomyocytes is 1000-fold lower than in fibroblasts/endothelial cells.
- Figure 3F: in my opinion figure S5 should be included here
- Figure 4C and 5C: is body weight stable between the two groups or does it affect the HW/BW. Preferably correct for tibia length, as it is less affected by the experimental procedure.

Text

• Page 8, last line of first paragraph: Echocardiographic analysis, moreover, revealed that WIF1 KO mice had developed more severe cardiac dysfunction... (to emphasize that the other group also has severe cardiac dysfunction).

1st Revision - authors' response

02 May 2017

Referee #1 (Remarks):

This manuscript explores the impact of the cardiac local microenvironment on the infarcted myocardium and monocyte activation. Specifically, the authors found that the Wnt inhibitory factor 1, WIF1, plays a key role during monocyte activation by controlling the inflammatory process after myocardial infarction (MI). RNAseq analysis shows that non-canonical WNT signaling is increased in Ly6Chi monocytes isolated from the heart, but not from the blood or bone marrow, in mice following MI, reflecting the microenvironment's interaction with recruited monocytes. In vitro analysis revealed a strong upregulation of WIF1 in isolated neonatal cardiomyocytes - but not fibroblasts or endothelial cells - upon hypoxia. In vivo, WIF1 was found to be elevated at early time points after MI (days 1 and 3 post-MI). The important role of WIF1 is underscored by the work done of WIF1 KO mice. Compared to WT, WIF1 KO mice showed impaired cardiac function marked by increased scar size and reduced ejection fraction four weeks after MI and the heart of WIF1 KO contained significantly more inflammatory monocytes than those of WT mice. Further exploring the modulatory role of WIF1, the authors induced AAV-mediated cardiomyocyte-specific WIF1 overexpression, which attenuated the monocyte response and improved cardiac function after MI, as compared to control-AAV-treated animals. Finally, WIF1 overexpression in isolated neonatal cardiomyocytes limited the activation of non-canonical WNT signaling and led to reduced IL1 β and IL6 expression in monocytes/macrophages. In light of their results, the authors suggest WIF1 is a key player in restricting the local inflammatory monocyte response, and thereby exhibits cardioprotective properties.

Overall, this study reveals interesting findings with potentially important clinical relevance. While the data are sound, they are not always clearly presented and they suffer from significant weaknesses that need to be addressed along the following points:

We thank the reviewer for acknowledging the importance of our findings and the soundness of our data. We have carefully reevaluated our manuscript and believe that we were able to address the important concerns of the reviewers.

Major concerns:

1- In contrast to previous reports suggesting that WNT antagonists (frizzled-related protein 5 and DKK3) decreased following MI, the authors found elevated WIF1 during the inflammatory phase of myocardial healing (Figure 3). However, a recent study showed that 4 days post-MI a downregulation (2.2-fold) of WIF1 was observed in the heart (Palevski et al, J Am Heart Assoc. 2017;6:e004387. DOI: 1). The same group also found a downregulation of GSK3 β , while in the current manuscript an up-regulation of GSK3 β was observed early following MI. The authors should clarify this issue.

To specifically address this discrepancy, we reevaluated our data, performed additional Western analysis and believe that the observed difference can be explained by differences in sampling of tissue. While from our understanding of their publication, Palevski et al. investigated whole heart tissue, we separated the cardiac infarct area from the border zone. Western blot analysis from tissue lysates revealed a down regulation of WIF1 compared to heart tissue from sham operated (in line with the data from Palevski et al.), but also showed a significant upregulation of WIF1 expression in tissue from the border zone, supporting our data shown in Figure 3C and D.

Western blot analysis of WIF1 protein expression in borderzone, infarct area and sham operated animals. Left: representative western blot. Right: Quantification of WIF1 expression at different locations. (normalized to sham, n=3, *p<0.05).

To clarify these observed differences, we have specified our sampling in the methods section (page 19, line 24) and figure legend 2 and added the following point of discussion: “Other groups have found WIF1 to be downregulated on an mRNA level post MI. In contrast, we observed an increase in WIF1 protein levels which might be explained by different sampling of whole heart vs. borderzone tissue.” (page 13, line 7)

We apologize for the insufficient description that might have led to a confusion regarding GSK3 β . Palevski et al. demonstrated a downregulation of GSK3 β in whole heart tissue after MI. We did not evaluate how GSK3 β expression changes in the heart of infarcted and sham-operated animals. However, we observed in our transcriptome analysis that monocytes isolated from the heart express GSK3 β to a higher extent compared to monocytes in the bone marrow (Figure 1D). From these observations we cannot conclude how GSK3 β expression changes in whole heart tissue. We have revised the figure legend to improve the understanding accordingly:

“Figure 1. Differential gene expression profiles in inflammatory monocytes sorted from the bone marrow, blood and heart were found three days after MI. A) RNA-seq analyses revealed differential expression of 1482 genes in monocytes sorted from different bodily regions. B) PANTHER Pathway analysis of genes found in the transcriptomes. C) Differential gene expression of WNT-associated genes in monocytes. D) Log₂(x-fold) of canonical WNT pathway inhibitors in Ly6C^{hi} monocytes sorted from the heart compared to Ly6C^{hi} monocytes in the bone marrow. E) Log₂(x-fold) of non-canonical WNT/PCP pathway mediators in Ly6C^{hi} monocytes sorted from the heart compared to Ly6C^{hi} monocytes in the bone marrow. “

2-To validate the role of the non-canonical WNT/PCP pathway, the authors evaluated phosphorylated JNK (pJNK) expression levels in post-MI murine heart whole-tissue lysates and they found increased pJNK levels until day 7 post-MI as compared to sham-operated animals. In figure 2 A and B, the data are normalized to actin expression. However, the potential changes are highly dependent on the extent of the induced infarct. The data in general should be first normalized to the cardiac Troponin T (cTnT) levels monitored on day 1 after MI. This holds also for figures 3-5.

We thank the reviewer for the suggestions on normalizing to the infarct size. We agree that potential changes can greatly depend on the extent of the induced infarct. However, we are not sure how to normalize to cTnT when comparing sham-operated to MI animals (figure 2 and 3). There is no cTnT detectable (<10pg/ml) in the blood of sham-operated animals, which makes normalizing to cTnT impossible. Unfortunately, we were not able to find any publication that uses cTnT-levels for normalization for reference.

When we compared two MI Groups to each other (WIF1KO vs WT or WIF1-AAV vs WT, figure 4 and 5) we monitored cTnT levels on day one after LAD ligation and believe that is an important and relevant quality control in this mouse model. However, we did not observe differences in the initial infarct sizes (i.e. cTnT levels) between the two groups. We therefore consider the infarct sizes equal between the groups and the observed differences should be

explained by the Genotype and subsequent alterations in healing and not due to the extent of induced infarcts.

3- In Figure 3F, an immunostaining of WIF1 is provided with tissue sections from human patients with acute MI. A control tissue section of WIF1 should be provided from an individual not suffering from MI or a related cardiac disease.

We were able to obtain control tissue sections from patients who were not suffering from cardiac related disease (e.g. death due to trauma). Staining for WIF1 did not reveal any relevant expression in these sections. We have added this new information to new Figure 3F and updated the figure legend accordingly (changes in red).

4-In Figure 6, supernatant of hypoxic cardiomyocytes activated JNK phosphorylation in isolated monocytes/macrophages. This activation was significantly reduced after transfer of supernatant from hypoxic cells that overexpressed WIF1. To validate the impact of WIF1 on non-canonical WNT signaling in accumulating monocytes, did the authors check the MMP13 levels in isolated monocytes/macrophages?

To validate the impact of WIF1 on non-canonical WNT signaling in accumulating monocytes, we have performed additional experiments including the infarction of WIF1KO and WT animals, isolation of accumulating monocytes on day 3 after MI and advanced the panel of non-canonical WNT markers. This analysis supported the finding that the absence of WIF1 increases non-canonical Wnt signaling in these leukocytes. We have added this information to the revised manuscript (new Supplemental Figure 7):

“Analysis of monocytes isolated on day 3 after LAD ligation from the hearts of WIF1 KO animals showed increased transcription levels of the downstream target of the AP-1/cJun transcription factor MMP13 and components of the non-canonical WNT signaling pathway ROR2, Rhoa, Rhou, Daam1, Dvl2 compared to monocytes isolated from wild type animals (sup. fig. 7). These findings may indicate that WIF1 impacts non-canonical WNT signaling in accumulating monocytes in vivo.”

S7) Expression of non-canonical WNT components in monocytes isolated from the heart of WT and WIF1KO animals three days after MI.

5- The potential paracrine impact of ischemic cardiomyocytes via WIF1 on local monocytes, should be also discussed in light of the possible effects on the niche of cardiac progenitor cell self-renewal.

We thank the reviewer for this suggestion. We added the following paragraph to the discussion to address this aspect.

“We here describe the paracrine effect of cardiomyocyte-secreted WIF1 on monocytes. Other celltypes such as cardiac progenitor cells (CPCs) might also be influenced by WIF1 as it has been reported that activation of WNT signaling can interfere the self-renewal of adult CPCs and blocks cardiac regeneration ².”

Minor concerns:

1-The supplementary figures should be re-numbered, according to the order of presentation of the data in the results section. For example, at the beginning of the results, supplementary fig. 6 is presented, why it is not numbered as Suppl. Fig 1.

We have rearranged the figure numbers according to the reviewer’s suggestion.

2-While described in the text, no supplementary tables (1 and 2) were provided in the files.

We apologize for this mistake. Please find the tables in the revised manuscript.

3- The title of the legend of figure 3 should be: "WIF1 expression during MI: and not "WIF1 protein expression during MI", because both mRNA and protein data are provided.

We thank the reviewer for pointing this out. We changed the figure legend of figure 3 from “WIF1 protein expression during MI” to “WIF1 expression during MI” as suggested.

4- The data on MMP13 in monocytes (p.9) are not quoting the supplementary fig.8 that describes them.

Thank you for the remark. We are now quoting supplementary figure 7 (formerly sup fig. 8) in the mainmanuscript.

Referee #2 (Remarks):

This is a well-written and clearly structured manuscript describing the identification of non-canonical WNT pathway modulators in the ischemic heart with a role in shaping the inflammatory and reparative response of the infiltrating monocytes. It argues for a specific role of the cardiac microenvironment (specifically the injured cardiomyocytes themselves) in controlling monocyte activity via the WNT pathway. Specifically, the non-canonical WNT modulator WIF1 was uncovered as a key player in this context. Notably, the causal role of WIF1 was shown by global KO and AAV-based cardiac overexpression. As controlling the inflammatory response in the first days after MI is viewed as a promising avenue to prevent adverse remodeling and heart function insufficiencies following MI, and although the current study is not (yet) fully convincing at this point, the data offer promise for novel anti-inflammatory strategies for ischemic cardiac injury. The WNT pathway is a molecular axis that had not been considered in this context. The paper therefore offers a good degree of novelty. However, some of the conclusions (in particular those related to JNK) are only partly supported by the data shown and some of the figures are of poor/insufficient quality to justify the conclusions. Overall, the molecular link between non-canonical WNT signaling and the JNK pathway is not yet convincingly shown.

We thank the reviewer for the constructive and very helpful feedback. We hope to adequately address the concerns and remove the ambiguities.

The following major criticism should be addressed:

Where the RNAseq hits confirmed by qPCR?

Sorting of inflammatory monocytes from the bone marrow and the heart was performed to reevaluate our findings regarding WNT signaling (i.e. cFos, cJun, LRP1 and Dab2) by qPCR. These analyses confirmed the major RNAseq hits. The results are depicted in Supplemental Figure 1:

S1) Confirmation of major RNAseq hits in monocytes isolated from the BM compared to monocytes isolated from the heart by qPCR.

It is surprising that typical non-canonical WNT regulators like Sfrp5 or WNT5a are not showing up in the transcriptome analysis. How can this be explained?

Sfrp5 and WNT5a have been shown to be crucial molecules after cardiac injury and also impact healing after MI³. Our analysis focused on infiltrating monocytes which strongly respond to local WNT proteins rather than releasing them (at least at these early time points of inflammation, as suggested by our experiments involving bone marrow transplantation, Supplemental Figure 8). Other cell types, especially cardiomyocytes might therefore be the main producer of WNT regulators. In addition, WNT components are known to be very potent, but their expression is often low. Expression levels of e.g. sfrp5 or WNT5a may therefore not reach the detection threshold in our RNAseq analysis.

What ROR2, a typical non-canonical WNT receptor?

To address this question, we analyzed ROR2 expression on monocytes in the bone marrow and the heart three days after induction of MI by flow cytometry. Comparison of ROR2 fluorescent intensity revealed an upregulation in heart compared to bone marrow monocytes. These data were added to the revised manuscript (new Supplemental Figure 2) and are integrated into the results section (page 7, line 1):

“In addition, FACS analysis of ROR2, a key receptor for non-canonical Wnt signaling was found to be upregulated in heart compared to bone marrow monocytes (Supplemental Figure 2).”

Figure legend Supplementary Figure 2: Flow cytometric analysis of inflammatory monocytes demonstrate an increased expression of ROR2 in the heart compared to bone marrow monocytes.

Furthermore, we see increased ROR2 mRNA expression in monocytes isolated from the heart of WIF1KO mice compared to heart monocytes of wildtype mice (please see the revised supplementary figure. 7 and our response to your comment below).

The meaning of Suppl. Fig. S6 is not really clear.

Processing heart tissue for FACS-sorting requires digestion of the tissue to obtain a single cell suspension. This procedure step is not necessary for the processing of the bone marrow. In order to exclude the possibility that digestion may lead to transcriptional alterations, we performed an additional cell sorting in which we treated both the heart as well as the bone marrow cells with the digestion mixture. We then performed qPCR analysis on the major hits of the RNAseq and were able to rule out a relevant impact of the isolation procedure and confirmed the observed major RNAseq hits. We have expanded the figure legend of supplementary figure 1 (formerly supplementary figure 6) to further clarify this aspect:

“Confirmation of major RNAseq hits in monocytes isolated from the BM compared to monocytes isolated from the heart by qPCR. Major RNAseq hits of the canonical WNT pathway (*Lrp1*, *Dab2*) and the non-canonical WNT pathway (*Fos*, *Jun*) could be confirmed by qPCR with a similar X-fold increase.”

A main weakness of the paper is that other upstream pathways (receptors) that can lead to JNK activation in the ischemic heart are not considered at all (TNF, CD74 signaling etc.). These should be ruled out by specific blocking experiments to underpin the role of the WNT pathway.

We thank the reviewer for this very important comment. We fully agree that TNF α signaling is often a key aspect in the activation of JNK. As suggested by the reviewer, we therefore performed blocking experiments with anti-TNF α and anti-CD74 IgG antibodies and respective isotype controls. Regarding anti-TNF α antibodies: we incubated supernatant of cardiomyocytes that were cultured under hypoxic or normoxic conditions with 5-10 μ g/ml anti-TNF α antibody (R&D and abcam) or with their IgG controls respectively for 1-4h. We then incubated isolated monocytes with the antibody-treated supernatant.

Regarding anti-CD74 antibody: We isolated monocytes and incubated them with 10 μ g/ml anti-CD74-antibody (abcam) or with its IgG control for 1h. We then incubated the treated monocytes with supernatant of cardiomyocytes that were cultured under hypoxic or normoxic conditions.

However, these antibodies failed to reduce pJNK expression in our *in vitro* setup. We therefore measured TNF α -levels in the supernatant of cardiomyocytes that were cultured under hypoxic or normoxic conditions respectively using ELISA (R&D Systems). TNF α -levels did not reach the detection limit. We then concentrated the supernatant (~20x fold) using Amicon Ultracel -10K centrifugal filters (Millipore). There was no difference in TNF α -levels in concentrated supernatant of cardiomyocytes cultured under hypoxic or normoxic conditions. In addition, the maximum measured concentration of TNF α in the concentrated Supernatant was 47 pg/ml, which gives a maximum concentration of 2,35 pg/ml TNF α in the supernatant we use for stimulation of monocytes *in vitro* (see below). These TNF α levels appeared to be insufficient for JNK activation in our hands.

TNF α levels in concentrated supernatant of cardiomyocytes cultured under hypoxic and normoxic conditions (left) and extrapolated TNF α levels in supernatants that were used for monocyte stimulation (right).

These experiments however do not rule out the possible impact of TNF α on JNK activation *in vivo*. We therefore added the following to our revised manuscript: “*In vitro*, we found that hypoxic cardiomyocyte supernatant activates the non-canonical WNT pathway in macrophages through JNK and ATF2 phosphorylation. By contrast, the canonical WNT pathway was downregulated in the same macrophages, a result that corroborated the findings described above (fig. 2C and D and sup. fig. S3). Another key mediator of JNK phosphorylation is TNF α . We could not detect significant differences in TNF α levels (data not shown) between supernatant of cardiomyocytes cultured under hypoxic and normoxic conditions indicating that TNF α might not be the driving force of JNK phosphorylation in our set-up.”

And in the discussion:

“Accordingly, non-canonical WNT signaling is activated in whole-heart tissue during the first week after MI, a phase characterized by the presence of leukocytes in the heart. In addition, our *in vitro* data suggest that troubled cardiomyocytes can activate non-canonical WNT signaling in monocytes directly, since monocyte/macrophage stimulation with hypoxic cardiomyocyte supernatant led to increased JNK and ATF2 phosphorylation and simultaneously decreased canonical WNT pathway. TNF α is known to mediate JNK phosphorylation and was found to be upregulated in heart tissue following acute MI⁴. Although we were not able to detect differences in TNF α in the supernatant of cardiomyocytes cultured under normoxic and hypoxic conditions, we cannot exclude a contribution of TNF α signaling or other upstream pathways in non-canonical WNT/JNK signaling in monocytes/macrophages *in vivo*.”

Also, while generally well-written with a nice flow for the reader to follow, the transition from Figure 1D/E to JNK as a main WNT-driven culprit pathway is hard to follow for the reader. Actually, the entire JNK part is a bit confusing. The expression of total JNK is discussed in Figure

1E, but Figure 2 then somehow switches to pJNK. Total JNK levels usually are rather stable, but seem to be upregulated as part of non-canonical WNT activation in the heart.

We fully agree with the reviewer that JNK levels are rather stable, but have indeed found an upregulation of JNK (on a transcriptional level) between monocytes sorted from the heart compared to monocytes sorted from the bone marrow. In order to improve readability and clarify the switch to pJNK, we have modified the manuscript:

“Phosphorylation of JNK is a crucial step in the activation of the WNT/PCP pathway. To further examine the role of the non-canonical WNT/PCP pathway, we evaluated phosphorylated JNK (pJNK) expression levels in post-MI murine heart whole-tissue lysates.”

Figure 2 is unconvincing. First of all, it is strange that only one JNK band (which one: 46 or 54 kD?, full blot for Figure 2A?) shows up. In contrast, in Figure 6A, 2 JNK bands can be seen. Molecular weight markers are missing. Secondly, what about total JNK. If pJNK as a measure of short-term JNK activation is measured, total JNK levels must be used for standardization rather than beta-actin. Also, the quality of the blots (and bands is rather poor).

In order to address this concern, we have repeated the complete experiment and normalized on total JNK according to the reviewers' suggestion (resulting in revised figures 2 and 6). Upregulation of pJNK remained significantly upregulated 2 days after MI after normalization to total JNK rendered significant. We believe that the quality of the blots has been greatly improved.

Figure 2

Figure 2. Non-canonical WNT increases following MI. A) Representative western blots of pJNK expression in the border zone of mouse hearts following MI. B) Quantification of pJNK

expression (mean±SD, N=5, *P≤0.01). C) Representative western blots and (D) quantification of pATF2, pJNK and active beta catenin (ABC) expression in macrophages stimulated with supernatant of cardiomyocytes cultured under hypoxic conditions (mean±SD, N=5, *P≤0.05)

Along the same lines, other targets than MMP13 should be measured to establish the link between WIF1 and non-canonical WNT signaling in the accumulating monocytes.

To validate the impact of WIF1 on non-canonical WNT signaling, we expanded the panel of non-canonical WNT markers measured in accumulating monocytes of WIF1KO and WT animals after MI. This resulted in a revised supplementary figure 7:

S7) Expression of non canonical WNT components in monocytes isolated from the heart of WT and WIF1KO animals three days after MI.

The relevant paragraph was revised as followed:

“Analysis of monocytes isolated on day 3 after LAD ligation from the hearts of WIF1 KO animals showed increased transcription levels of the downstream target of the AP-1/cJun transcription factor MMP13 and components of the non-canonical WNT signaling pathway ROR2, Rhoa, Rhou, Daam1, Dvl2 compared to monocytes isolated from wild type animals (sup. fig. 7). These findings may indicate that WIF1 impacts non-canonical WNT signaling in accumulating monocytes in vivo.”

Minor:

Figure 3E: size bars are missing.
We added the missing scale bars.

2. It is unclear why a pie diagram was used in Figure S7? How many mice do these percentage numbers refer to? I cannot find this number?

We believe that the pie diagram illustrates the differences in mortality in our case quite well. A total of 106 mice were included in this analysis (52 in the group of knockout animals and 54 in the group of C57BL6 wild type animals). (The pie diagram is now listed as supplementary figure 6)

3. The BMT experiment from WIF1 KO mice is not well explained. What else was measured? Was there an impact on infarct size and/or heart function? What are the conclusions?

Our key read out in this experiment was the analysis of monocyte numbers in the infarcted heart. FACS analysis revealed no difference between both groups. Following this result, we did not measure infarct size or cardiac function in these groups of mice, as we concluded that the main source of WIF1 must be other cells than infiltrating immune cells. We revised the manuscript as shown below:

“To evaluate whether the WIF1 in myeloid cells itself impacts the immune response following MI, we performed bone marrow transfers from WIF1 KO mice into WT recipients. FACS analysis of hearts on day four after MI showed no significant differences between the groups’ numbers of Ly6C^{hi} monocytes and Ly6C^{lo} macrophages (sup. fig. S8). We therefore concluded that non-myeloid cells rather than infiltrating immune cells are the main source of WIF1 in the heart after ischemic insult.”

Referee #3 (Remarks):

The authors provide an interesting view on the role of Wnt signalling during myocardial infarction. Their differentiation between the canonical and non-canonical signalling can be very relevant for the interpretation of previous and ongoing studies. The manuscript is well written. The authors provide data in support of their conclusions. Especially the evidence regarding cardiomyocytes as main source of WIF1 is supported by clear in vitro and in vivo data. However, there are some critical issues that need to be addressed before a final conclusion can be drawn.

General comments

- RNA sequencing provides an unbiased approach to identify transcriptional changes. However, the method highly depends on sample preparation and input, and the authors describe how monocytes have been sorted by FACS. Therefore, it is surprising to find MYH6, a highly specific marker for cardiomyocytes, among the enriched genes in heart-derived monocytes (figure 1C). The same holds to a lesser extent for Colla1, which is also highly expressed by myocytes and fibroblasts. It would be good if the authors reflect on this and provide additional evidence that the sorting was performed properly.

This is a very important comment and we took great effort to be as precise in our cell isolation procedure as possible. We have performed a rigorous purity check and added this information to the supplemental material (new Supplemental Figure 11).

S11) Purity control for sorted inflammatory monocytes.

While we cannot fully rule out the possibility of contamination with individual cells, the differences in the size of cardiomyocytes and fibroblasts compared to monocytes should have made it highly unlikely, as these cells were excluded based on the scatterplots right away. One speculative explanation regarding MYH6 was that phagocytosis of cell debris might influence the result. For Colla1 an expression in monocytes has been previously described⁵ and might be an early sign of Ly6C high monocytes differentiating into fibrocytes.

- In line with the previous comment: it can be argued that there are many other factors, besides the myocardial infarction, that underlie the changes in gene expression when comparing monocytes from bone marrow, blood, and heart. I am aware that this is probably the only feasible approach, but suggest that the authors discuss this limitation.

We agree that this approach has its limitations, but we also believe that it is highly relevant and the best approach feasible at the moment. We have discussed the limitation in the revised manuscript (page 11, line 24):

“Changes in gene expression in monocytes might also occur due to the transition from bone marrow to blood and heart also beyond the impact of myocardial infarction. A clear limitation in this regard is that sorting inflammatory monocytes from sham-operated hearts for RNAseq is not feasible.”

- The authors ascribe the changes in cardiac pJNK and pATF2 to the infiltrating monocyte fraction (figure 2). JNK and ATF2 expression is however not restricted to monocytes, and it is unlikely that the observed effect can only be ascribed to infiltrating monocytes.

We fully agree that other cell types beyond monocytes might show non-canonical WNT activation. To discuss this aspect we changed the manuscript as followed:

“We found more components of the non-canonical WNT pathway and intracellular canonical WNT pathway inhibitors in monocytes isolated from the heart than in monocytes from the bone marrow. These data suggest that the non-canonical WNT pathway may be activated in monocytes in the infarcted heart. Accordingly, non-canonical WNT signaling is activated in whole-heart tissue during the first week after MI, a phase characterized by the presence of leukocytes in the heart. Despite these findings, one has to take in to account that other cell types such as cardiomyocytes or fibroblasts may contribute to the observed activation of non-canonical WNT signaling in whole heart tissue.”

- The conclusion that effects of WIF1 inhibition result from changes in MMP13 expression is quite strong, and it would be good to provide some additional evidence for changes in ECM remodelling (figure 1C, for example, actually shows upregulation of Colla1 expression, which would speak for the contrary).

We apologize for the misunderstanding. We do not conclude that the observed effects result from changes in MMP13. We used MMP13 as a maker/target gene of non-canonical WNT. We think that increased non-canonical WNT signaling in monocytes leads to a stronger inflammatory response (a stronger pro-inflammatory phenotype of the accumulating monocytes) and subsequently additional tissue damage. To further clarify this aspect we have expanded the panel of non-canonical WNT markers and rephrased the result section as followed:

“Analysis of monocytes isolated on day 3 after LAD ligation from the hearts of WIF1 KO animals showed increased transcription levels of the downstream target of the AP-1/cJun transcription factor MMP13 and components of the non-canonical WNT signaling pathway ROR2, Rhoa, Rhou, Daam1, Dvl2 compared to monocytes isolated from wild type animals (sup. fig. 7). These findings may indicate that WIF1 impacts non-canonical WNT signaling in accumulating monocytes in vivo.”

S7) Expression of non canonical WNT components in monocytes isolated from the heart of WT and WIF1KO animals three days after MI.

- The authors fail to put their observations in a broader perspective. For example: WIF1 has been shown to play a role in early angiogenesis (Melgar-Lesmes et al. PMID: 26022689), and WNT-signalling regulates myofibroblast recruitment (Blyszczuk et al. PMID: 27099262). The discussion would greatly benefit from a broader review of (contradicting) evidence.

We thank the reviewer for this suggestion and have added the following paragraph to the discussion in order to put our observations in a broader perspective (page 14, line 19).

“We here describe the paracrine effect of cardiomyocyte-secreted WIF1 on monocytes. Other celltypes such as cardiac progenitor cells (CPCs) might also be influenced by WIF1 as it has been reported that activation of WNT signaling can interfere the self-renewal of adult CPCs and blocks cardiac regeneration ².

WNT signaling has also been reported to play an important role in myofibroblasts formation and fibrosis in cardiac diseases ^{6 7}. Blyszczuk et al and Duan et al found that inhibition of WNT signaling limits fibrosis and may be beneficial during the healing process following myocarditis and myocardial infarction.

Interestingly, Melgar-Lesmes and Edelman found that infiltrating monocytes colocalize with non-canonical WNT protein WNT5a following partial hepatectomy and may support vascular growth during liver regeneration ⁸. Inhibition of WNT signaling could therefore also lead to adverse effects regarding neovascularization following MI. These findings show the complexity of WNT signaling and the importance of understanding WNT signaling in a spatial-temporal manner. “

Figures

- Figure 1: provide an overview of the gating strategy, eventually as supplemental figure. See comments above.

We have added the information on our gating strategy into a new Supplementary Figure 12:

S12) Gating strategy for cell sorting of inflammatory monocytes.

• Figure 2A,C and Figure 6A: It can be argued that pJNK and pATF2 represent the active and relevant read-out, but it would be very informing to show western blots of total JNK and ATF2 levels.

Please also see our comment above. We repeated a complete set of experiments including a time-line to address this issue and updated figure 2 and figure 6 accordingly.

We furthermore moved the phosphorylation of ATF2 to figure 6. The activation of ATF2 measured with ATF2 reporter assay can still be found in sup. fig. 3. Please find the revised figure 2 and 6 below.

Figure 2

Figure 2. Non-canonical WNT increases following MI. A) Representative western blots of pJNK expression in mouse hearts following MI. B) Quantification of pJNK expression (mean \pm SD, N=5, *P \leq 0.05). C) Representative western blots and (D) quantification of pATF2, pJNK and active beta catenin (ABC) expression in macrophages stimulated with supernatant of cardiomyocytes cultured under hypoxic conditions (mean \pm SD, N=5, *P \leq 0.05).

Figure 6. WIF1 inhibits non-canonical WNT signaling. A) Scheme of *in vitro* experiments B) mRNA levels of inflammatory markers in macrophages stimulated with supernatant of control or WIF1 overexpressing cardiomyocytes cultured under hypoxic conditions (mean±SD, N=3, *P≤0.05). C) Representative western blots of JNK and ATF2 expression in macrophages stimulated with supernatant of AdControl- or AdWIF1-transfected hypoxic cardiomyocytes and D) Quantification of pJNK and pATF2 expression in macrophages stimulated with supernatant of AdWIF1-transfected hypoxic cardiomyocytes (mean±SD, N=5, *P≤0.05).

• Figure 3A-C: Is there any information regarding "absolute" levels of expression of WIF1 in the cell types? The conclusions are affected if its expression in cardiomyocytes is 1000-fold lower than in fibroblasts/endothelial cells.

While we do not have absolute expression levels, analysis of WIF1 by qPCR with different cell types rendered significantly lower Ct values in cardiomyocytes compared to fibroblasts, indicating a higher abundance of WIF1 in cardiomyocytes.

Figure 3F: in my opinion figure S5 should be included here

We were able to obtain heart tissue sections from patients free from cardiac-related diseases. The staining is now included in figure 3F please find below the revised figure 3F:

• Figure 4C and 5C: is body weight stable between the two groups or does it affect the HW/BW. Preferably correct for tibia length, as it is less affected by the experimental procedure.

We have analyzed the data according to the reviewers suggestion and found a significant difference in body weight of WIF1KO animals and their WT littermates (see below on the left). Body weight between AAV-WIF1 and the control group did not differ significantly (see below on the right).

We therefore and calculated the heart weight / tibia length ratio and adapted figure 4C accordingly. The difference between WT and WIF1KO remained significant.

“4C) Heart weight/tibia length ratios four weeks after induced MI (mean±SD, N=11, *P≤0.01”

• Page 8, last line of first paragraph: Echocardiographic analysis, moreover, revealed that WIF1 KO mice had developed more severe cardiac dysfunction... (to emphasize that the other group also has severe cardiac dysfunction).

Thank you for the suggestion. We changed the sentence accordingly.

1. Palevski D, Levin-Kotler LP, Kain D, Naftali-Shani N, Landa N, Ben-Mordechai T, Konfino T, Holbova R, Molotski N, Rosin-Arbesfeld R, Lang RA, Leor J. Loss of macrophage wnt secretion improves remodeling and function after myocardial infarction in mice. *J Am Heart Assoc.* 2017;6:004387
2. Oikonomopoulos A, Sereti KI, Conyers F, Bauer M, Liao A, Guan J, Crapps D, Han JK, Dong H, Bayomy AF, Fine GC, Westerman K, Biechele TL, Moon RT, Force T, Liao R. Wnt signaling exerts an antiproliferative effect on adult cardiac progenitor cells through igfbp3. *Circ Res.* 2011;109:1363-1374
3. Nakamura K, Sano S, Fuster JJ, Kikuchi R, Shimizu I, Ohshima K, Katanasaka Y, Ouchi N, Wash K. Secreted frizzled-related protein 5 diminishes cardiac inflammation and protects the heart from ischemia reperfusion injury. *J Biol Chem.* 2015
4. Jacobs M, Staufenberger S, Gergs U, Meuter K, Brandstatter K, Hafner M, Ertl G, Schorb W. Tumor necrosis factor-alpha at acute myocardial infarction in rats and effects on cardiac fibroblasts. *J Mol Cell Cardiol.* 1999;31:1949-1959
5. Crane MJ, Daley JM, van Houtte O, Brancato SK, Henry WL, Jr., Albina JE. The monocyte to macrophage transition in the murine sterile wound. *PLoS One.* 2014;9
6. Duan J, Gherghe C, Liu D, Hamlett E, Srikantha L, Rodgers L, Regan JN, Rojas M, Willis M, Leask A, Majesky M, Deb A. Wnt1/betacatenin injury response activates the epicardium and cardiac fibroblasts to promote cardiac repair. *Embo J.* 2012;31:429-442
7. Blyszczuk P, Muller-Edenborn B, Valenta T, Osto E, Stellato M, Behnke S, Glatz K, Basler K, Luscher TF, Distler O, Eriksson U, Kania G. Transforming growth factor-beta-dependent wnt secretion controls myofibroblast formation and myocardial fibrosis progression in experimental autoimmune myocarditis. *Eur Heart J.* 2016;20
8. Melgar-Lesmes P, Edelman ER. Monocyte-endothelial cell interactions in the regulation of vascular sprouting and liver regeneration in mouse. *J Hepatol.* 2015;63:917-925

2nd Editorial Decision

02 June 2017

Thank you for the submission of your revised manuscript to EMBO Molecular Medicine. We have now received the enclosed reports from the 2 out of the three reviewers that were asked to re-assess it. Unfortunately I failed to obtain a re-evaluation from reviewer 3. As a further delay cannot be justified, I am proceeding with the two available evaluations. As you will see reviewers 1 and 2 are now globally supportive. As for reviewer 3, we have now considered your rebuttal at the editorial level, and found your actions and replies to be satisfactory and to fully address his/her concerns.

I am therefore pleased to inform you that we will be able to accept your manuscript pending the

following final amendments:

- 1) Please reformat the in-text citations and reference list according to our guidelines (<http://embomolmed.embopress.org/authorguide#referencesformat>)
- 2) Please provide supplementary information formatted as per our guidelines (<http://embomolmed.embopress.org/authorguide#expandedview>). Supplementary information must be presented as a singly PDF file beginning with a short table of contents. Also, appendix items should be refereed to in the manuscript as Appendix Figure S1, Appendix Table S1, Appendix Supplementary Methods.
- 3) Please add a size bar to figure S4
- 4) As per our Author Guidelines, the description of all reported data that includes statistical testing must state the name of the statistical test used to generate error bars and P values, the number (n) of independent experiments underlying each data point (not replicate measures of one sample), and ALL actual P values for each test (not merely 'significant' or 'P < 0.05').
- 5) Every published paper now includes a 'Synopsis' to further enhance discoverability. Synopses are displayed on the journal webpage and are freely accessible to all readers. They include a short standfirst as well as 2-5 one sentence bullet points that summarise the paper. Please provide the synopsis including the short list of bullet points that summarise the key NEW findings. The bullet points should be designed to be complementary to the abstract - i.e. not repeat the same text. We encourage inclusion of key acronyms and quantitative information. Please use the passive voice. Please attach this information in a separate file or send them by email, we will incorporate it accordingly. You are also welcome to suggest a striking image or visual abstract to illustrate your article. If you do please provide a jpeg file 550 px-wide x 400-px high.
- 6) We are now encouraging the publication of source data, particularly for electrophoretic gels and blots, with the aim of making primary data more accessible and transparent to the reader. Would you be willing to provide a PDF file per figure that contains the original, uncropped and unprocessed scans of all or at least the key gels used in the manuscript? The PDF files should be labeled with the appropriate figure/panel number, and should have molecular weight markers; further annotation may be useful but is not essential. The PDF files will be published online with the article as supplementary "Source Data" files. If you have any questions regarding this just contact me.

Please submit your revised manuscript within two weeks. I look forward to seeing a revised form of your manuscript as soon as possible.

***** Reviewer's comments *****

Referee #1 (Remarks):

The authors have adequately addressed my concerns and revised the manuscript, which improved its quality.

Referee #2 (Remarks):

This reviewer's criticism has been addressed appropriately.

Corresponding Author Name: Florian Leuschner
Journal Submitted to: EMBO Molecular Medicine
Manuscript Number: EMM-2017-07565